JCB | Journal of Cell Biology

# Molecular basis of MKLP2-dependent Aurora B transport from chromatin to the anaphase central spindle

Michela Serena , Ricardo Nunes Bastos , Paul R. Elliott , and Francis A. Barr

The Aurora B chromosomal passenger complex (CPC) is a conserved regulator of mitosis. Its functions require localization first to the chromosome arms and then centromeres in mitosis and subsequently the central spindle in anaphase. Here, we analyze the requirements for core CPC subunits, survivin and INCENP, and the mitotic kinesin-like protein 2 (MKLP2) in targeting to these distinct localizations. Centromere recruitment of the CPC requires interaction of survivin with histone H3 phosphorylated at threonine 3, and we provide a complete structure of this assembly. Furthermore, we show that the INCENP RRKKRR-motif is required for both centromeric localization of the CPC in metaphase and MKLP2-dependent transport in anaphase. MKLP2 and DNA bind competitively to this motif, and INCENP T59 phosphorylation acts as a switch preventing MKLP2 binding in metaphase. In anaphase, CPC binding promotes the microtubule-dependent ATPase activity of MKLP2. These results explain how centromere targeting of the CPC in mitosis is coupled to its movement to the central spindle in anaphase.

## Introduction

Aurora B is a crucial mitotic kinase that regulates chromosome condensation in prophase, microtubule attachment to kinetochores during prometaphase and metaphase, and anaphase spindle microtubule dynamics and cytokinesis (Carmena et al., 2012). These functions require localization to specific sites on chromosome arms, centromeres, and microtubules (Hindriksen et al., 2017). Aurora B localization and activation throughout mitosis and cytokinesis depend on three other proteins: INCENP, survivin, and borealin (Adams et al., 2000; 2001; Gassmann et al., 2004; Romano et al., 2003; Sampath et al., 2004; Vader et al., 2006; Wheatley et al., 2001). Together, these proteins form the tetrameric chromosomal passenger complex (CPC). The CPC localizes to the chromosome arms in prophase and then becomes enriched on centromeres during prometaphase and metaphase. During mitotic exit, it is relocated to the anaphase central spindle and then destroyed as cells enter G1 (Cooke et al., 1987). In addition to mediating localization, the interaction of Aurora B with the C-terminal IN-box sequence within INCENP stabilizes the active form of the kinase (Sessa et al., 2005). Hence, Aurora B activation and localization are coordinated.

A fundamental insight into the molecular mechanism underpinning the complex spatial and temporal pattern of CPC localization came from the identification of a ternary subcomplex formed by survivin, borealin, and the N-terminal

region of INCENP (Klein et al., 2006). Further structural analysis defined a 58–amino acid sequence at the N terminus of INCENP that forms a minimal trimeric complex with survivin and the first 76 amino acids of borealin (Jeyaprakash et al., 2007). Within this assembly, survivin plays a key role in recognition of centromeric chromatin by binding to a specific phosphorylated chromatin mark on histone H3. This histone H3 phosphothreonine 3 (H3pT3) mark is created by the protein kinase haspin and is temporally restricted to mitosis since haspin is activated by CDK1–cyclin B (Ghenoiu et al., 2013; Kelly et al., 2010; Wang et al., 2010; Yamagishi et al., 2010). During mitotic exit, PP1-repoman, a counteracting phosphatase inhibited by CDK1–cyclin B, then dephosphorylates H3pT3 (Qian et al., 2011; Trinkle-Mulcahy et al., 2006; Vagnarelli et al., 2011). However, rather than explaining the release of the CPC from centromeres and the movement to the central spindle in anaphase, histone H3 dephosphorylation by this pathway appears be important for chromatin decondensation late during mitotic exit (Qian et al., 2015; Vagnarelli et al., 2006). This suggests that additional factors must contribute to and regulate CPC targeting.

Accordingly, although the ternary subcomplex (survivin, borealin[1-76], and the minimal module comprising the first 58 amino acids of INCENP [INCENP[1-58]]) has been proposed to be the minimal assembly required for targeting to centromeric chromatin in mitosis, CPC targeting is further enhanced by

---

Department of Biochemistry, University of Oxford, Oxford, UK.

Correspondence to Francis A. Barr: francis.barr@bioch.ox.ac.uk;   R. Nunes Bastos's present address is ONI, Linacre House, Oxford, UK.



additional signals in both INCENP and borealin. The INCENP single α-helical domain is involved in chromatin and microtubule binding (Samejima et al., 2015; Wheelock et al., 2017). Borealin dimerizes through a structured domain at the C terminus (Bekier et al., 2015; Bourhis et al., 2009) and makes direct and specific contact to nucleosomes (Abad et al., 2019). These properties will increase the avidity of the CPC for chromatin. Centromere-specific enrichment in mitosis is promoted by CDK1 phosphorylation of an unstructured region of borealin upstream of the dimerization domain, which promotes interaction with the centromeric protein shugoshin (Tsukahara et al., 2010). Phase separation is emerging as a key organizing principle for chromatin architecture, including the centromere (Gibson et al., 2019; Trivedi et al., 2019). The unstructured central region of borealin exhibits liquid–liquid phase separation in vitro and plays an important role in CPC self-organization and targeting to the centromere (Trivedi et al., 2019). This multivalent interaction network is thought to explain the selective binding of the CPC first to chromosome arms and then its enrichment at centromeres.

The complexity of this interaction network raises questions about the mechanism promoting removal of the CPC from chromatin and transport to the central spindle in anaphase. This appears to be an active process involving the mitotic kinesin MKLP2 and protein dephosphorylation. In cells depleted for MKLP2, Aurora B and the other CPC components remain on separating chromosomes during anaphase and fail to relocate to the central spindle (Gruneberg et al., 2004). Conversely, in cells depleted for CPC components, MKLP2 fails to move to the central spindle in anaphase (Hümmer and Mayer, 2009). This process is inhibited in metaphase cells by CDK1–cyclin B-dependent phosphorylation of INCENP at T59 and MKLP2 at multiple sites (Hümmer and Mayer, 2009; Kitagawa et al., 2014). These sites are dephosphorylated by PP2A-B55 in mitotic exit, enabling relocation of the entire CPC to the anaphase spindle (Hein et al., 2017). How MKLP2 recognizes the centromeric pool of the CPC and promotes its release from chromatin remains unclear. We therefore set out to address this question that is central to understanding the mechanism of Aurora B transport from chromatin to the anaphase spindle.

## Results

### CPC targeting to centromeres and the central spindle requires the first 80 amino acids of INCENP

The CPC makes multiple contacts with chromatin, and MKLP2 may compete for one or more of the interactions with phosphorylated histone H3 or nucleosomes. The search for the MKLP2-binding region on the CPC can be narrowed down to the N-terminal region of INCENP for two reasons (Fig. 1 A). First, phosphorylation at T59 has been shown to prevent CPC transport (Hümmer and Mayer, 2009). Second, the first 68 amino acids of INCENP have been reported to support localization to the anaphase spindle (Ainsztein et al., 1998). By contrast, INCENP$^{1-58}$ together with endogenous full-length borealin and survivin localizes to chromatin in mitotic cells, but it fails to support CPC targeting to the central spindle in anaphase (Klein et al., 2006).

To identify the INCENP anaphase-targeting signal, we created stable cell lines expressing inducible copies of GFP-INCENP at levels comparable with the endogenous protein (Fig. 1 B). Because of the requirement for Aurora B in mitotic chromosome and spindle formation, we first performed an analysis of localization in these cell lines in the presence of endogenous INCENP. This approach revealed that the first 80 amino acids of INCENP are required for the central spindle localization in anaphase (Fig. 1, C and E). We then reexamined the targeting of INCENP in metaphase cells by using either transient transfection as in previous studies or the stable cell lines. Using transient expression, we found that INCENP$^{1-58}$ targeted weakly to chromosomes in <35% of cells, whereas INCENP$^{1-80}$ showed chromatin targeting in all cells observed (Fig. S1 A). In the stable cell lines where expression is matched to the endogenous INCENP, INCENP$^{1-58}$ targeted weakly to chromosomes with a large cytoplasmic signal, whereas INCENP$^{1-80}$ showed centromere targeting and an undetectable cytoplasmic signal (Fig. S1 B; and Fig. 1, D and F). Together, these data indicate that INCENP contains an additional signal between amino acids 59 and 80 required for robust targeting to both the chromatin in metaphase and the central spindle in anaphase. Inspection of the N-terminal sequence of INCENP for conserved motifs revealed the presence of a basic region at amino acids 63–70 downstream of the CDK phosphorylation site thought to prevent CPC transport in metaphase (Fig. 1 A). Mutation of this RRKKRR-motif to a series of alanine residues abolished the targeting of INCENP$^{1-80}$ to both the anaphase central spindle (Fig. 1, C and E) and centromeres (Fig. 1, D and F). By contrast, phospho-mimetic mutation of T59 to glutamate (T59E) prevented CPC targeting to the anaphase spindle (Fig. 1, C and E) but had no effect on centromere targeting (Fig. 1, D and F). Similar behavior was observed for all the different GFP-INCENP mutants in cells depleted of endogenous INCENP (Fig. S1, C–E). The INCENP RRKKRR-motif mutant remained on chromatin and MKLP2 transport to the anaphase spindle was also perturbed (Fig. S1 D, RRKKRR). The INCENP RRKKRR-motif may therefore form part of a CDK-regulated signal required for central spindle targeting in anaphase and also play a role in robust chromatin targeting in metaphase.

### H3pT3 recognition by the CPC

We then sought to understand how this RRKKRR-motif–targeting signal cooperates with the known H3pT3 binding properties of the CPC. The structure of the N-terminal 58 amino acids of INCENP in complex with survivin and a minimal fragment of borealin from amino acids 10–109 has been determined previously (Jeyaprakash et al., 2007). However, the authors of this study were unable to model borealin beyond residue 76, owing to lack of interpretable electron density. Hereafter, we refer to this structure as chromosomal passenger with INCENP$^{NT}$ (CPI$^{NT}$). In addition, the structure of dimeric survivin bound to an H3T3-phosphorylated peptide has been solved (Du et al., 2012; Jeyaprakash et al., 2011; Niedzialkowska et al., 2012). However, the structure of the CPI$^{NT}$ bound to H3pT3 has not been determined. Although our attempts to cocrystallize CPI$^{NT}$ with the H3pT3 peptide also failed, we were successful in obtaining crystals of CPI$^{NT}$, without bound ligand (Table S1). These

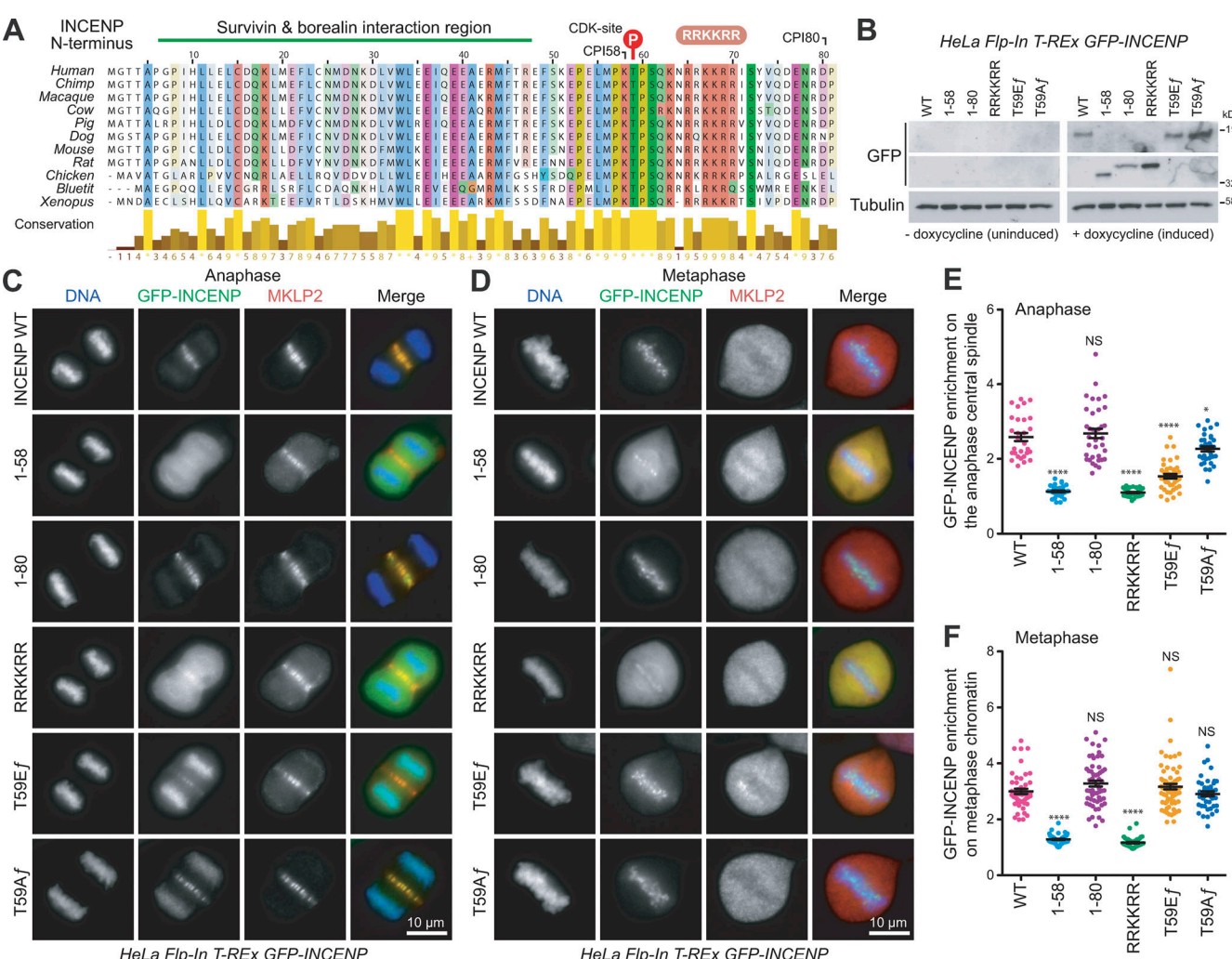

Figure 1. **Determinants for CPC localization in to chromosomes and the anaphase spindle. (A)** A ClustalX alignment of INCENP sequences from different species highlighting the survivin–borealin interaction region, the CDK phosphorylation site at T59, and the conserved RRKKRR-motif. **(B)** HeLa Flp-In T-Rex cells, expressing different doxycycline-inducible GFP-INCENP constructs (full-length [$f$] WT, T59E $f$, or T59A $f$ point mutants or truncations 1–58, 1–80, RRKKRR (1–80 RRKKRR to hexa-alanine mutant) were Western blotted before and after doxycycline induction. The INCENP antibody recognizes an epitope in the C terminus and does not detect the N-terminal fragments. **(C and D)** The induced cells were stained for MKLP2, and representative images of thymidine-treated cells in either anaphase (C) or metaphase (D) are shown. **(E and F)** GFP-INCENP enrichment ($f_{central\_spindle}/f_{cytoplasmic}$) at the central spindle (WT $n$ = 29, 1–58 $n$ = 36, 1–80 $n$ = 36, RRKKRR $n$ = 31, T59E $f$ $n$ = 39, T59A $f$ $n$ = 34; E) or metaphase chromatin ($f_{chromatin}/f_{cytoplasmic}$; WT $n$ = 50, 1–58 $n$ = 50, 1–80 $n$ = 58, RRKKRR $n$ = 51, T59E $f$ $n$ = 72, T59A $f$ $n$ = 43; F) are plotted in the graphs. Both graphs show the mean with individual data points marked, and error bars indicate SEM. An unpaired $t$ test with Welch's correction and 99% confidence intervals was performed (*, $P < 0.05$; ****, $P < 0.0001$).

structures revealed that in the crystals, CPI[NT] dimerized due to crystal packing, an effect observed, but not commented on, in previous work (Jeyaprakash et al., 2007). The N terminus of INCENP from a symmetry-related CPI[NT] molecule projects into the peptide-binding site of survivin (Fig. S2 A) with the same position and orientation as a phospho-histone H3 peptide bound to dimeric survivin (Du et al., 2012; Jeyaprakash et al., 2011; Niedzialkowska et al., 2012). This would displace any bound ligand and explain why previous attempts to obtain structures for CPI[NT] phospho-histone complexes failed.

Using this information, we removed the first six amino acids of INCENP and succeeded in obtaining crystals and solving structures for both INCENP[7-58] (Fig. 2 A, CPI58) and INCENP[7-80] (Fig. 2 B, CPI80) in complex with full-length survivin and borealin[10-76] bound to a 12–amino acid phosphopeptide derived

from the N terminus of histone H3 (H3pT3: ART[P]KQTARKSTG). Both complexes crystallized in the $P65$ space group and diffracted to 1.81 and 2.55 Å, respectively. We were able to identify the first five residues of the bound peptide (A1–Q5) for CPI58 and the first seven residues (A1–A7) for CPI80. However, we were unable to see INCENP from 45 onward, suggesting it might be flexible, as reasonably expected around the T59 CDK1 phosphorylation site. In both CPI58 and CPI80 complexes, H3pT3 was bound to survivin that, as previously described, is stabilized by a structural $Zn^{2+}$ ion, coordinated by C57, C60, H77, and C84 (Du et al., 2012; Niedzialkowska et al., 2012).

In both CPI58 and CPI80, the H3pT3 peptide is positioned in a negatively charged groove on the surface of the baculovirus inhibitor of apoptosis repeat domain of survivin (amino acids 15–89; Fig. 2, A and B), with the phosphorylated T3 directed

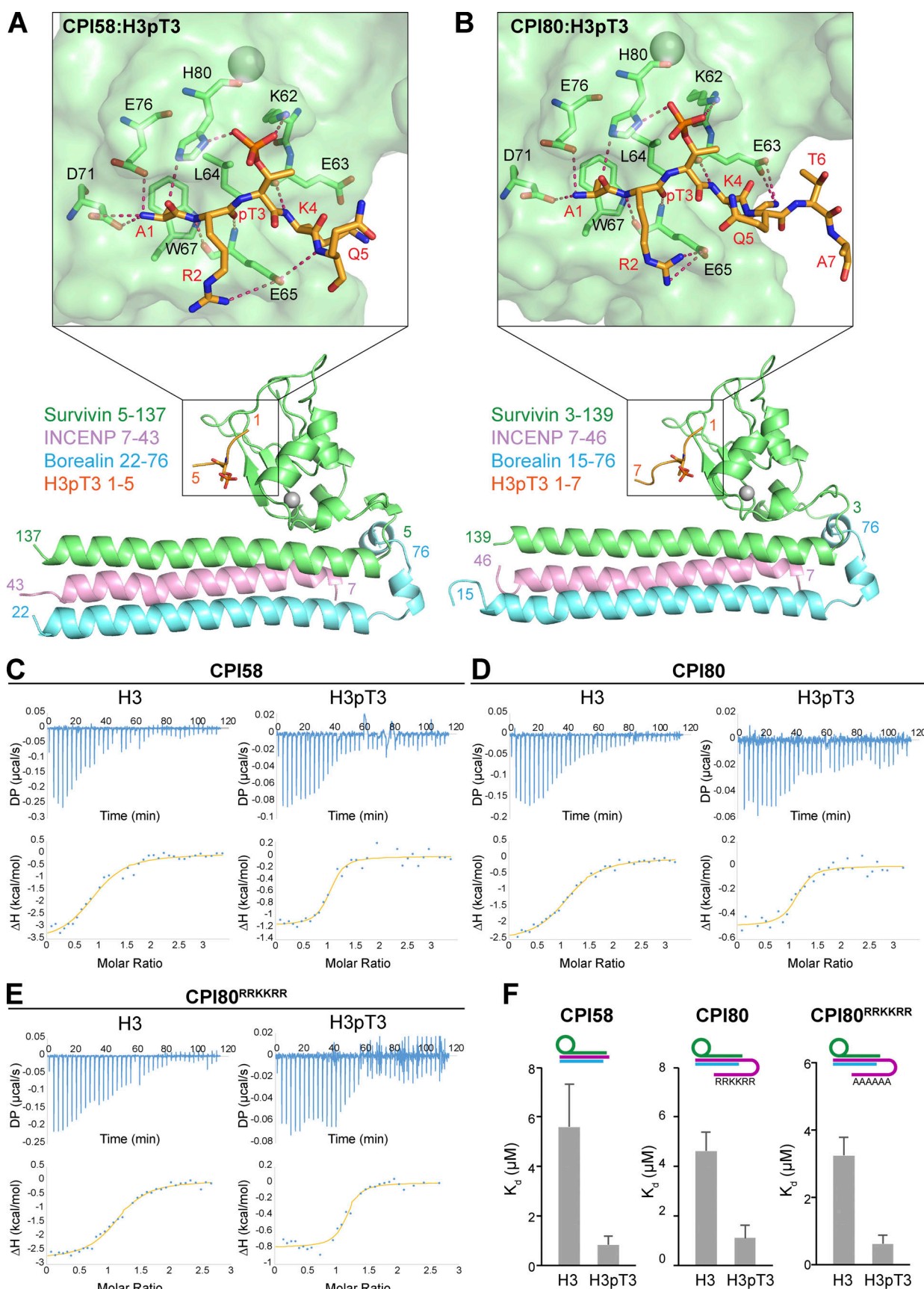

Figure 2. **Structure of the CPC bound to T3-phosphorylated histone H3. (A and B)** Crystal structures of CPI58 (chromosomal passenger with INCENP[7-58]; A) and CPI80 (chromosomal passenger with INCENP[7-80]; B) in the presence of the H3pT3 peptide (survivin in green, borealin in light blue, INCENP in pink, and

H3pT3 in orange) are shown. The close-up view highlights the intermolecular interactions between the BIR domain of survivin and H3pT3 (amino acids 1–5) for CPI58 and survivin and H3pT3 (amino acids 1–7) for CPI80. The free N-terminal A1 of H3pT3 contacts the negatively charged side chains of survivin D71 and E76. Hydrogen bonds also mediate interactions between amino acids A1, R2, and K4 of the H3pT3 peptide and amino acids H80, E65, and E63 of survivin, respectively. Recognition of phospho-threonine 3 in H3pT3 peptide is mediated by amino acids K62 and H80 of survivin. Hydrogen bonds are indicated with red dashed lines; survivin amino acids are labeled in black, and peptide amino acids are labeled in red. **(C–E)** ITC binding curves for complex formation between CPI58 (C), CPI80 (D), and CPI80$^{RRKKRR}$ (E) and either H3 or H3pT3. **(F)** Histograms show $K_d$ values, and error bars represent the $K_d$ calculation uncertainty, obtained using MicroCal PEAQ-ITC analysis software. Cartoons represent the protein complexes used for each ITC experiment, where survivin is in green, borealin in blue, and INCENP in purple (1–58, 1–80, and 1–80 RRKKRR). ΔH, enthalpy; DP, differential power.

toward a positively charged region (Fig. S2, B and C, electrostatic surface). Phospho-T3 recognition is mediated by hydrogen bond formation with the survivin K62 side chain and H80. Interestingly, in line with previously reported data (Du et al., 2012), we identified a difference in the orientation of the side chain of survivin K62 in the presence of H3pT3, compared with unliganded CPI$^{NT}$ (PDB accession no. 2QFA), confirming a role for K62 in phosphate recognition (Fig. S2, B and C). The histone H3 peptide is positioned within the binding pocket through a series of interactions that explain the sequence recognition and selectivity for the N terminus. The side chain of the free N-terminal Ala1 is inserted in a small hydrophobic pocket formed by survivin L64 and W67 (Fig. 2, A and B, enlarged regions). This limits the position of the N terminus of the peptide with respect to the phospho-binding pocket and explains the selectivity for phosphorylation at the 3-position of the peptide. Histone H3 peptide A1 is also involved in hydrogen bonds with survivin H80 and with the negatively charged side chains D71 and E76. Additional hydrogen bonds between R2 and K4 of the histone peptide and survivin E65 and E63, respectively, play a key role in positioning the signal in the binding pocket. For CPI58, we identified an additional hydrogen bond between the peptide backbone at Q5 and the survivin E65 side chain (Fig. 2 A) that was absent in the CPI80 structure (Fig. 2 B).

Comparison of the binding affinity for phosphorylated and nonphosphorylated histone H3 by CPI58 and CPI80 was then performed by isothermal titration calorimetry (ITC; Fig. 2, C and D). Both complexes bind to H3pT3 phosphopeptide with a relatively low dissociation constant ($K_d$) of ~1 µM (0.8 ± 0.3 µM for CPI58 and 1.1 ± 0.5 µM for CPI80) and to the H3T3 non-phosphopeptide with a $K_d$ of 4–6 µM (5.6 ± 1.7 µM for CPI58 and 4.6 ± 0.7 µM CPI80; Fig. 2 F). This interaction was lost for CPI58$^{E65A-H80A}$ (Fig. S2 D) and CPI80$^{E65A-H80A}$ complexes (Fig. S2 E) missing critical survivin residues conferring histone H3 recognition, where no measurable binding constant could be determined for the H3 and H3pT3 peptides. Previous investigation of the interaction of survivin with phosphorylated peptides has not revealed clear selectivity for phosphorylated histone H3 compared with other phosphorylated proteins (Du et al., 2012; Jeyaprakash et al., 2011). We therefore analyzed binding of CPI58 and CPI80 to a second centromere-enriched chromatin mark: histone H2A phospho-threonine 121 (H2ApT121; Yamagishi et al., 2010). Both complexes failed to bind to either the unphosphorylated H2A sequence or the H2ApT121 phosphopeptide, where the phosphate was again at the 3-position of the peptide (Fig. S2, F and G), supporting the view that there is selectivity for histone H3. In summary, our data show that both CPI58 and CPI80 selectively associate with histone H3, with a moderate preference of pH3T3 over H3T3.

We then tested whether the INCENP RRKKRR-motif is required for binding to H3pT3. This region was not resolved in the CPI80:H3pT3 structure, suggesting it is flexible and fails to adopt a defined conformation. CPI80$^{RRKKRR}$ mutant complexes were produced where the RRKKRR-motif was replaced with alanine residues. ITC showed that CPI80$^{RRKKRR}$ had binding affinities comparable with WT CPI80 for both phosphorylated and nonphosphorylated histone H3, with a $K_d$ of 0.6 ± 0.3 and 3.3 ± 0.5 µM, respectively (Fig. 2 E), suggesting that this region of the protein does not contribute to binding of H3pT3.

These data confirm that the phosphopeptide-binding specificity of the CPC is conferred by survivin as previously reported. They also provide support for the idea that there is an additional signal for chromatin and centromere targeting, in addition to phospho-histone H3. We therefore investigated the role of the RRKKRR-motif in centromere targeting.

### The INCENP RRKKRR-motif mediates DNA binding

Because of the basic nature of the RRKKRR-motif, we hypothesized that it must interact with an acidic binding partner, and DNA was the most obvious candidate. To test whether the CPC interacts directly with DNA through the INCENP RRKKRR-motif, we performed electrophoretic mobility shift assays (EMSAs) with purified components. Addition of CPI80 to assays resulted in a mobility shift in the 2.9-kb linear duplex DNA used (Fig. 3 A). This shift was dose dependent in the micromolar range, between 2 and 3 µM protein (Fig. 3 B). By contrast, the CPI80$^{RRKKRR}$ mutant, CPI58 lacking the RRKKRR-motif, an unrelated negative control protein, or buffer alone did not cause this DNA mobility shift (Fig. 3 A). These findings were extended using microscale thermophoresis (MST) to measure CPC binding to specific centromeric α-satellite or genomic non–α-satellite DNA 40-bp duplexes (complete datasets are in Table S2). We used these sequences since most centromeres and neocentromeres are characterized by the presence of adenine thymidine (AT)-rich DNA, of which α-satellite is one example (Naughton and Gilbert, 2020). CPI80 showed a slightly higher affinity for centromeric α-satellite than non–α-satellite DNA, with $K_d$ values of 23.6 ± 1.6 and 29.7 ± 1.6 µM, respectively (Fig. 3 D). Both CPI58, which is truncated before the RRKKRR-motif (Fig. 3 E), and the CPI80$^{RRKKRR}$ mutant (Fig. 3 F) failed to show any specific interaction with DNA, and it was not possible to obtain values for $K_d$. In comparison with CPI80, CPI80$^{T59E}$, which mimics the CDK-phosphorylated form of the CPC found in prometaphase and metaphase, showed a twofold reduction in affinity for both centromeric α-satellite and non–α-satellite DNA (Fig. 3 G).

Together, these data support the conclusion that the interaction of the CPC with chromatin is mediated by both selective binding of histone H3 by survivin and interaction of the INCENP RRKKRR-motif with DNA. There is a slight ~1.25-fold but

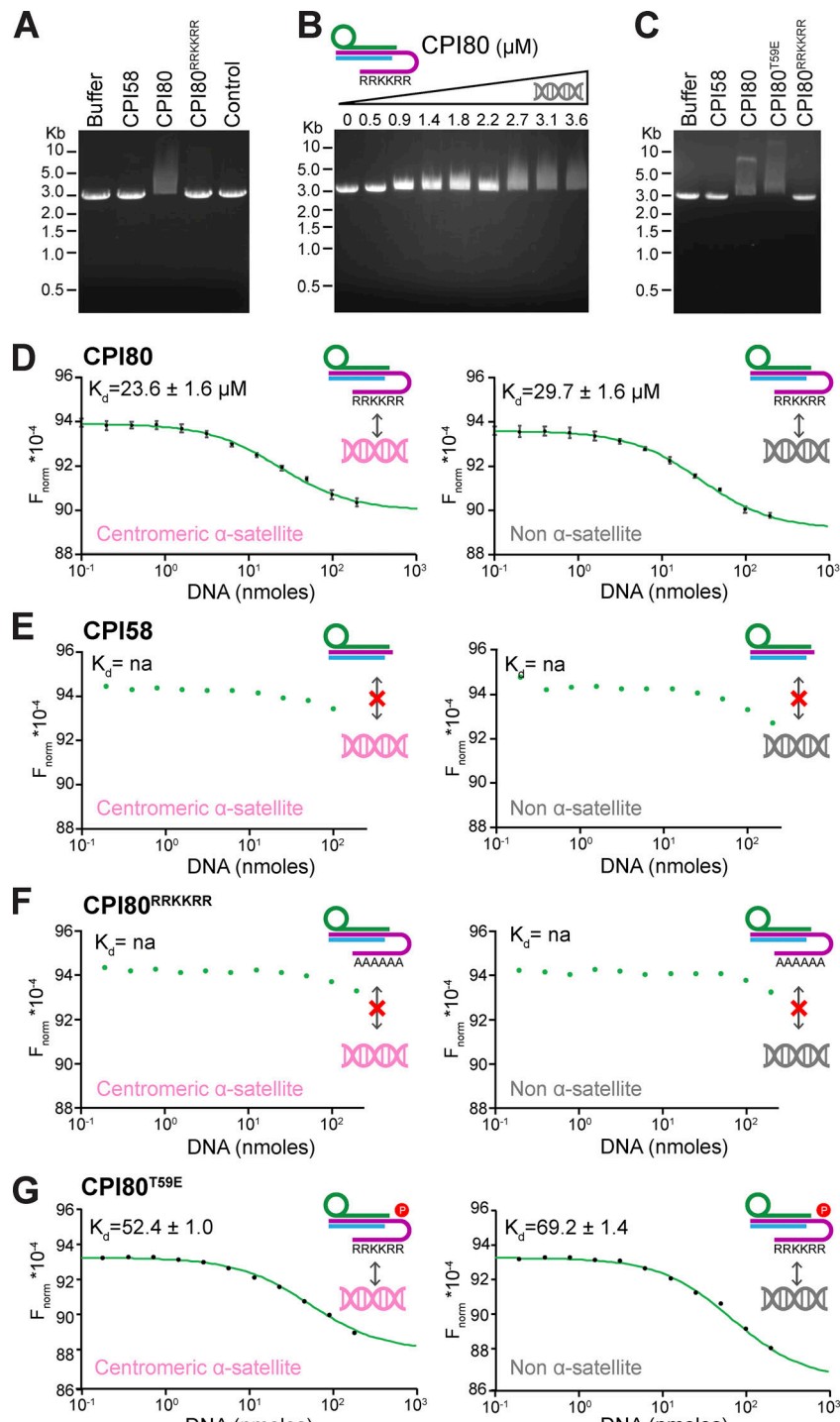

Figure 3. **CPC interacts with DNA through the IN-CENP RRKKRR-motif. (A)** EMSA with linear plasmid DNA incubated with 3.6 µM CPI58, CPI80, or CPI80$^{RRKKRR}$. Buffer indicates no protein was added, whereas control is a negative control protein in the same buffer that does not bind DNA (HRV 3C). **(B)** EMSA of linear plasmid DNA incubated with increasing concentrations of CPI80 from 0 to 3.6 µM. **(C)** EMSA of linear plasmid DNA incubated with 3.6 µM CPI58, CPI80, CPI80$^{T59E}$, or CPI80$^{RRKKRR}$ buffer. **(D–G)** MST of 0.1–200 µM CPI80 (D), CPI58 (E), CPI80$^{RRKKRR}$ (F), or CPI80$^{T59E}$ (G) with centromeric α-satellite (pink) or non–α-satellite (gray) DNA duplexes. $K_d$ was calculated for CPI80, where graph shows the mean and error bars indicate SEM ($n = 4$) using NanoTemper analysis software. A two-way ANOVA was performed, considering dilution factor and DNA sequence (centromeric α-satellite or non–α-satellite) as independent variables (****, $P < 0.0001$). It was not possible to calculate a $K_d$ value for the other conditions. Cartoons inset into the panels represent protein complexes used for each EMSA or MST experiment, where survivin is in green, borealin in blue, and INCENP in purple (1–80, 1–58, 1–80 RRKKRR, 1–80$^{T59E}$, respectively). F$_{norm}$, normalized fluorescence; na, not applicable.

reproducible preference for α-satellite DNA that suggests there may be some sequence specificity to the interaction, but this requires further investigation. We then investigated the relationship between the mechanisms of chromatin binding from prophase to metaphase and CPC release from chromatin and subsequent localization to the central spindle in anaphase.

**The INCENP RRKKRR-motif is also required for MKLP2 binding**
Because the INCENP RRKKRR-motif is also required for CPC targeting to the anaphase spindle, we initially asked whether it

is required for interaction with MKLP2. Immunoprecipitation of CPC complexes from synchronized metaphase and anaphase cells was used for this purpose. Full-length GFP-IN-CENP coprecipitated with the other CPC proteins borealin and survivin in metaphase and anaphase and showed specific temporal interaction with phosphorylated histone H3 in metaphase and MKLP2 in anaphase (Fig. 4, WT). The anaphase interaction with MKLP2 was not observed when a T59E phospho-mimetic form of INCENP was used (Fig. 4, T59E), consistent with the idea phosphorylation at this site by CDK1

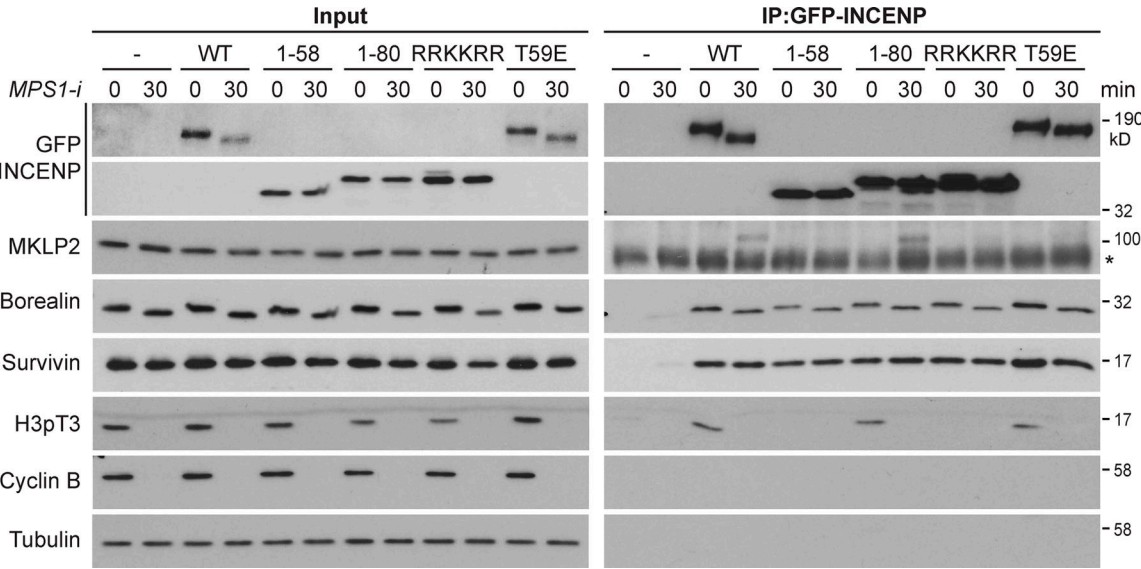

Figure 4. **RRKKRR-motif–dependent interaction of the CPC with H3pT3 in metaphase and MKLP2 in anaphase.** GFP-INCENP was immunoprecipitated (IP) with anti-GFP antibody from untransfected HeLa cells (horizontal dash) or HeLa cells transfected with GFP-INCENP WT, 1–58, 1–80, 1–80 RRKKRR, or T59E mutants. After nocodazole treatment, cells were collected at 0 and 30 min after the addition of the MPS1 inhibitor (*MPS1-i*; AZ3146). Immunoprecipitated protein complexes were analyzed by Western blotting. Asterisk marks a cross-reactivity with the antibody used for IP, detected in the MKLP2 blots of the IP samples.

prevents MKLP2-dependent transport in metaphase (Hümmer and Mayer, 2009). Importantly, INCENP$^{T59E}$ still coprecipitated phosphorylated histone H3 in metaphase (Fig. 4, T59E). INCENP$^{1-80}$ behaved in the same way as the WT full-length protein and showed specific temporal interaction with phosphorylated histone H3 in metaphase and MKLP2 in anaphase (Fig. 4, 1–80). In contrast, the INCENP$^{RRKKRR}$ mutant and INCENP$^{1-58}$ did not coprecipitate either phosphorylated histone H3 or MKLP2 (Fig. 4, RRKKRR and 1–58), in agreement with the localization data for these constructs (Fig. 1, C and D). MKLP2 interaction with the CPC in anaphase therefore requires the RRKKRR-motif that is also required for DNA binding and centromere targeting in metaphase. The loss of histone H3 interaction observed with the INCENP$^{RRKKRR}$ mutant and INCENP$^{1-58}$ is consistent with the data showing that this region is required for efficient localization to centromeres. We therefore conclude that two signals, binding to phosphorylated histone H3 by survivin and centromeric DNA binding by INCENP, cooperate to promote CPC localization to centromeres and that MKLP2 may compete for the DNA-binding site on INCENP. This latter observation may provide a mechanism to explain CPC release from chromatin in anaphase.

## MKLP2 binding to the CPC involves an acidic region

Having defined motifs on CPC required for efficient chromatin targeting and MKLP2 binding, we asked which regions of MKLP2 are required for interaction with the CPC. To address this question, we used a dominant-negative E413A ATPase-defective rigor mutant of MKLP2. When expressed in cells, this traps the CPC, followed using Aurora B, at an early step of the transport cycle at the boundary between chromatin and central spindle microtubules (Fig. S3 A). Progressive deletion of the C terminus of MKLP2 showed that once amino acids

C-terminal to position 650 are removed, Aurora B is released from this mutant MKLP2 trap. This maps the putative interaction site to a region between amino acids 650 and 750 (Fig. S3 A). Further fine truncation mapping defines a region close to amino acid 640–660 that is necessary to trap the CPC (Fig. S3 B). Overexpression of a fragment of MKLP2 containing this region was sufficient to block CPC transport to the anaphase spindle (Fig. S3 C), consistent with a previous report using a slightly larger fragment of MKLP2 (Kitagawa et al., 2014). Furthermore, this fragment of MKLP2 was coprecipitated with the CPC as cells progressed from metaphase into anaphase (Fig. S3 D), suggesting it competes for interactions with endogenous MKLP2. Sequence alignment revealed the presence of three conserved elements within this region (Fig. S4 A): an acidic patch from amino acids 636–652 that our structural analyses showed forms a paired coiled-coil elements (Fig. S4, B and C), an invariant RRSQR sequence from amino acids 675–679, and a weakly acidic element made up of amino acids 690–705. Mutation of the RRSQR sequence did not affect central spindle targeting of MKLP2, but it did reduce binding to and transport of the CPC compared with WT MKLP2 (Fig. 5, A and C). Mutation or deletion of these three elements showed that only the first acidic patch of amino acids 636–652 was essential for interaction of the CPC with MKLP2 in anaphase (Fig. 5 A). Because of the possible role for this region in MKLP2 dimerization, we confirmed the dimeric state of GFP-MKLP2 Δ636-652 by size exclusion chromatography (SEC). Both GFP-MKLP2 and the GFP-MKLP2 Δ636-652 deletion formed dimers indistinguishable from the endogenous WT protein (Fig. 5 B) and were estimated from calibration standards to have a size of ~200–250 kD. The loss of interaction of the CPC with GFP-MKLP2 Δ636-652 is therefore not an indirect consequence of a defect in dimerization. Consistent with the failure to coprecipitate with the CPC in anaphase, MKLP2 Δ636-652 was unable to support

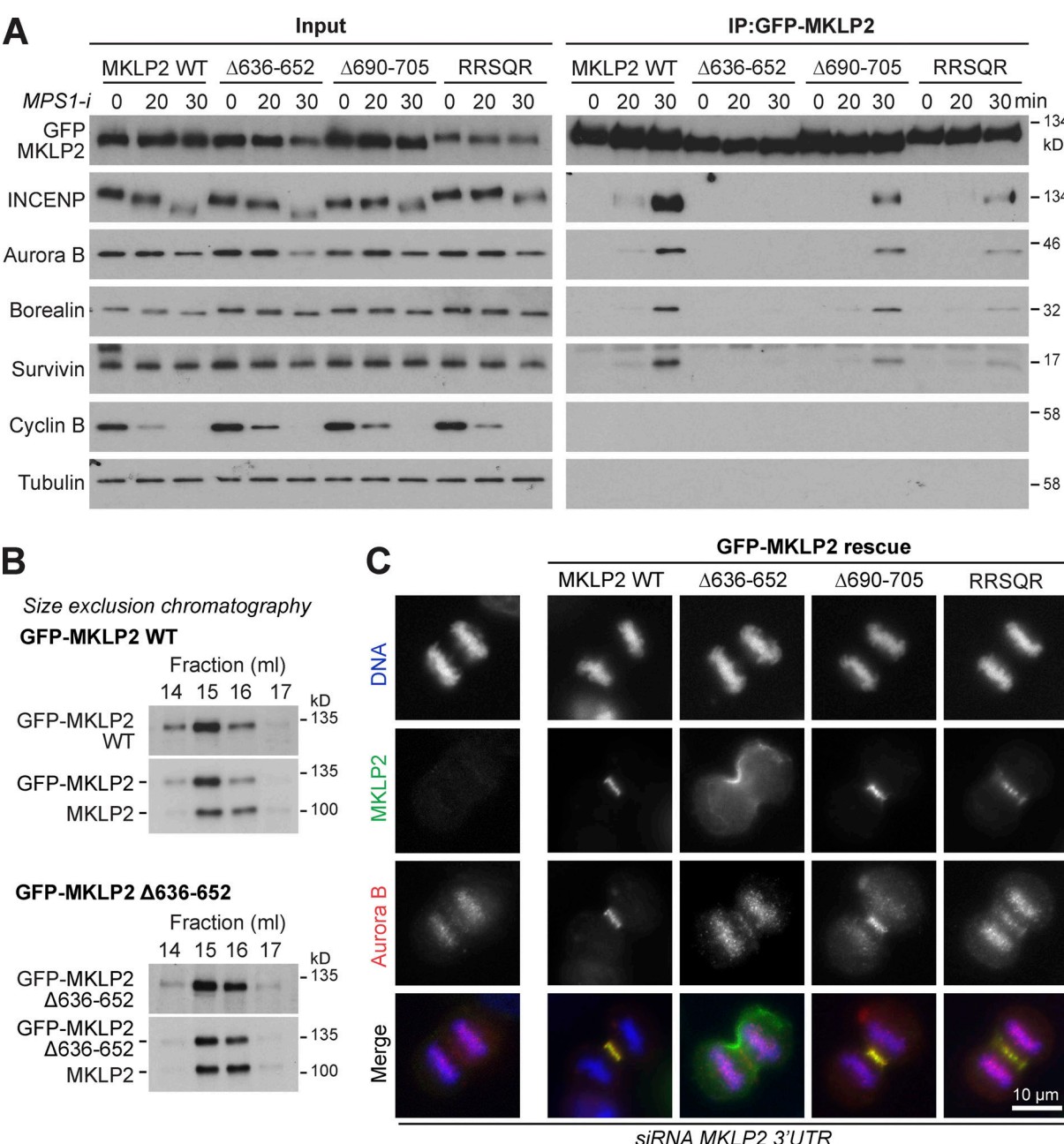

**Figure 5. MKLP2 amino acids 636–652 are necessary for interaction with the CPC in anaphase. (A)** GFP-MKLP2 was immunoprecipitated (IP) with anti-GFP antibody from HeLa cells transfected with GFP-MKLP2 WT or Δ636-652, Δ690-705, or RRSQR (penta-alanine substitution) mutants. After nocodazole treatment, cells were collected at 0, 20, and 30 min after the addition of Mps1 inhibitor (*MPS1-I*; AZ3146). Immunoprecipitated protein complexes were analyzed by Western blotting. **(B)** Comparison of GFP-MKLP2 and GFP-MKLP2 Δ636-652 to endogenous untagged MKLP2 using SEC. Peaks fractions of the elution were Western blotted for MKLP2 and GFP. **(C)** HeLa cells expressing GFP-MKLP2 WT or Δ636-652, Δ690-705, or RRSQR mutants were silenced for endogenous MKLP2 with a 3′ UTR siRNA and then stained for Aurora B. Representative images of cells in anaphase are shown.

localization of Aurora B to the central spindle (Figs. 5, A and C; and Fig. S4 D). Interestingly, MKLP2 Δ636-652 itself failed to localize to central spindle microtubules, in agreement with the idea that there is a mutual dependency for the CPC in MKLP2 activation and movement (Hümmer and Mayer, 2009).

**Reconstitution of the CPC–MKLP2 interaction**
We next sought to reconstitute the interaction of the CPC with MKLP2 using minimal purified components. SEC-multiangle

light scattering (MALS) was used to give an unbiased estimation of molecular mass and stoichiometry. Recombinant CPI80 forms a 1:1:1 trimer of survivin, borealin[10-109], and INCENP[1-80], with a molecular mass of 41.7 kD (Figs. 6 A and S4 E, orange traces), in agreement with the structural data showing a monomeric complex comprised of a single copy of each subunit. The minimal binding region of MKLP2[557-668] exists in two species with molecular masses of 25.4 and 63.7 kD (Figs. 6 A and S4 E, yellow traces), consistent with a dimeric and a higher order

oligomeric species. When mixed, CPI80 and MKLP2$^{557-668}$ form a single species with a molecular mass of 100.3 kD (Figs. 6 A and S4 E, blue traces), consistent with two copies of CPI80 bound to a dimer of MKLP2. This proposal agrees with the dimeric structure of the core MKLP2-binding region (Fig. S4, B and C), and dimeric structure for most known kinesin proteins (Hirokawa et al., 2009). By contrast, neither the CPI80$^{RRKKRR}$ mutant nor CPI58, which is truncated before the RRKKRR-motif, formed SEC-stable stoichiometric complexes with MKLP2$^{557-668}$, resulting in multiple peaks in the elution profile (Fig. 6, B and C, blue traces). This failure to stably bind MKLP2 thus explains why these truncated or mutated forms of INCENP do not support CPC localization to the central spindle in anaphase cells.

## Survivin-dependent chromatin targeting in metaphase is a prerequisite for CPC localization in anaphase

A key initial step in MKLP2-dependent transport of the CPC is its release from chromatin. We therefore asked whether chromatin targeting is essential for anaphase targeting of the CPC by creating survivin mutants that would disrupt chromatin binding, but not the interaction with MKLP2. To do this, we investigated the role of histone H3 binding by survivin by creating structure-guided mutations to differentiate the role of H3pT3 binding and H3 backbone binding in targeting to chromatin. Cells expressing these proteins were then depleted of endogenous survivin, and the expression level was confirmed by Western blotting (Fig. S5 A). Single survivin K62A and H80A mutants and a K62A-H80A double mutant that should disrupt phospho-threonine recognition (Fig. 2, A and B) showed apparently normal targeting to chromatin in metaphase (Fig. 7 A) and the central spindle in anaphase (Fig. 7 B). Careful examination of the images suggests that the survivin K62A-H80A mutant spreads away from centromeres and decorates the entire chromatin mass marked by a DNA dye (Fig. 7 A), consistent with the idea that H3pT3 plays a specific role in centromere targeting of the CPC (Niedzialkowska et al., 2012; Wang et al., 2010). Survivin E65 was then mutated alone or together with H80 to disrupt binding to histone H3 R2 and the peptide backbone adjacent to A1 (Figs. 2, A and B; and Fig. S2 E). In contrast to WT survivin and the other mutants tested, these survivin E65A or E65A-H80A double point mutants showed a diffuse cytosolic distribution in both metaphase (Fig. 7 A) and anaphase (Fig. 7 B). This effect was not due to loss of the MKLP2 interaction, since CPI80$^{Survivin\ E65A-H80A}$ was able to form SEC-stable complexes (Fig. 7 C) like the WT CPI80 (Fig. 6 A). Therefore, by uncoupling histone recognition from MKLP2 binding with the E65A mutant, we can show that survivin-dependent chromatin targeting in metaphase is a prerequisite for CPC localization in anaphase.

## MKLP2 competes with DNA for binding to the CPC

Using the MKLP2:CPI80 complexes, we then asked how MKLP2 promotes the release of the CPC from chromatin. From the data presented so far, there are two nonexclusive possibilities that we can test: MKLP2 competes with either phospho-histone binding or DNA binding. First, ITC was used to investigate the binding affinity of the MKLP2:CPI80 complex for phosphorylated and nonphosphorylated histone H3. The H3pT3 phosphopeptide

bound to the MKLP2:CPI80 complex with a relatively low $K_d$ of <1 μM (Fig. 8 A), similar to CPI80 or CPI58 in the absence of MKLP2 (Fig. 2 F). The nonphosphorylated H3 peptide bound to MKLP2:CPI80 with a $K_d$ of 2.8 ± 0.3 μM, a value slightly lower than that observed for CPI58 and CPI80 alone (Figs. 2 F and 8 A). These results show that MKLP2 does not compete for phospho-histone binding to the CPC.

Next, we tested whether MKLP2 binding competes for DNA binding to the CPC. MKLP2:CPI80 complexes failed to cause a pronounced mobility shift in EMSA assays (Fig. 8 B). By contrast, CPI80 addition gave rise to a large DNA mobility shift, whereas the CPI80$^{RRKKRR}$ mutant, CPI58 lacking the RRKKRR-motif, MKLP2$^{557-668}$, or buffer alone did not (Fig. 8 B). These findings were extended using MST assays to analyze the interaction of MKLP2:CPI80 complexes with specific centromeric α-satellite or non–α-satellite DNA sequences (Table S2). In contrast to CPI80 alone (Fig. 3 D), MKLP2:CPI80 failed to show any specific interaction with either centromeric α-satellite or non–α-satellite DNA, and it was not possible to obtain $K_d$ values (Fig. 8 C). We therefore conclude that MKLP2 promotes release of the CPC from chromatin by competing for the RRKKRR-motif–dependent interaction of INCENP with DNA.

## CPC binding stimulates MKLP2 ATPase activity and anaphase spindle localization

Finally, we asked whether the RRKKRR-motif is required for CPC transport and enrichment at the central spindle in anaphase cells. To do this, HeLa Flp-In T-REx GFP-INCENP or GFP-INCENP$^{RRKKRR}$ cells were used. The cells were depleted of endogenous INCENP with a 3′ UTR siRNA, and either full-length INCENP WT or the full-length INCENP RRKKRR mutant induced to endogenous levels (Fig. S5 B). In metaphase, INCENP localized to centromeres, whereas this was reduced for the INCENP RRKKRR mutant (Fig. S5, C and D). This was associated with reduced targeting of Aurora B to centromeres and a reduction, but not complete loss, of the Aurora B–dependent phosphorylation of histone H3 at S10 (Fig. S5, C and E). In anaphase, the RRKKRR mutant INCENP, defective for interaction with MKLP2, bound to chromatin but failed to localize to the central spindle (Fig. 9, A and B). MKLP2 also failed to localize to the central spindle and was spread out in patches at the cell cortex (Fig. 9, A and B).

Thus, the codependent transport of MKLP2 and the CPC is RRKKRR-motif dependent, and we conclude it requires direct interaction of the two protein complexes. As already noted, MKLP2 Δ636-652 deleted for the CPC-binding region fails to localize to central spindle microtubules, further supporting the conclusion that there is a mutual dependency for CPC in MKLP2 activation and movement. Furthermore, CDK-dependent phosphorylation of both the CPC on INCENP T59 and MKLP2 at multiple sites is thought to inhibit CPC transport and MKLP2 function in metaphase. To directly test these ideas, we performed a kinetic analysis of microtubule-stimulated ATPase activity for full-length MKLP2 incubated with different forms of the CPC. Mitotically phosphorylated MKLP2 has low microtubule-stimulated ATPase activity (Fig. 9 C, MKLP2P) that showed only a slight increase when incubated with CPI80 (Fig. 9 C, MKLP2P+CPI80). By contrast, nonphosphorylated MKLP2 has

Figure 6. **INCENP RRKKRR-motif dependant interaction of CPI80 with MKLP2. (A)** SEC-MALS chromatograms and molecular weight of MKLP2[557-668] (yellow), CPI80 (orange), and MKLP2:CPI80 (blue) are shown. MKLP2:CPI80 has a single peak and unique mass showing MKLP2 binds CPC in a 1:1 ratio. **(B and C)** SEC of MKLP2[557-668] (yellow) with CPI80[RRKKRR] (orange; B) or CPI58 (orange; C) was performed to test for MKLP2:CPI80[RRKKRR] or MKLP2:CPI58 (blue) complexes, respectively. Dotted lines indicate the fraction numbers. InstantBlue-stained acrylamide gels are shown for all proteins, where M, S, and B stand for MKLP2, survivin, and borealin, respectively. For INCENP, I[80] is amino acids 1–80, I[RRKKRR] is amino acids 1–80 with the RRKKRR to hexa-alanine mutation, and I[58] is amino acids 1–58. In the cartoon, depictions of protein complexes are as follows: survivin is in green, borealin in blue, INCENP in purple, and MKLP2 in yellow. mAU, milli-absorbance unit.

a markedly higher ATPase activity (Fig. 9 C, MKLP2) that increases approximately fourfold when CPI80 is added (Fig. 9 C, MKLP2+CPI80). CPI58[T59E] failed to stimulate the ATPase activity of MKLP2 (Fig. 9 C, MKLP2+CPI80), explaining why this is not transported to the anaphase spindle in cells. Similarly, CPI58 and CPI80[RRKKRR] did not increase the MKLP2 ATPase activity above the baseline observed with MKLP2 and microtubules alone (Fig. 9 D). We conclude that RRKKRR-motif–dependent and CDK-regulated binding of the CPC promotes MKLP2 ATPase activity and therefore facilitates the codependent transport of both partners to the anaphase central spindle.

## Discussion

### Moving toward a more complete understanding of CPC dynamics

MKLP2 actively promotes the movement of the CPC from chromatin to the central spindle at the onset of anaphase. Here, we

explain how MKLP2 promotes the release of the CPC from chromatin by competing for binding of DNA to the RRKKRR sequence in INCENP (Fig. 10). This mechanism explains why, in cells depleted of MKLP2, the CPC is retained on chromatin during anaphase rather than being released into the cytoplasm (Gruneberg et al., 2004). In the absence of MKLP2, the combined interaction of INCENP with DNA, survivin with histone H3, and borealin with nucleosomes is responsible for the retention of the CPC on chromatin (Abad et al., 2019; Du et al., 2012; Jeyaprakash et al., 2011; Niedzialkowska et al., 2012). This chelation effect, where multiple weak interactions help stabilize the CPC on chromatin, allows both tight binding to specific regions of the chromosome and dynamic release at the onset of anaphase. The tight temporal coupling of the release and transport process to the metaphase-to-anaphase transition can be explained by the CDK1–cyclin B-dependent phosphorylation of INCENP at T59 (Hümmer and Mayer, 2009; Kitagawa et al., 2014), which lies directly adjacent to the RRKKRR-motif needed for interaction

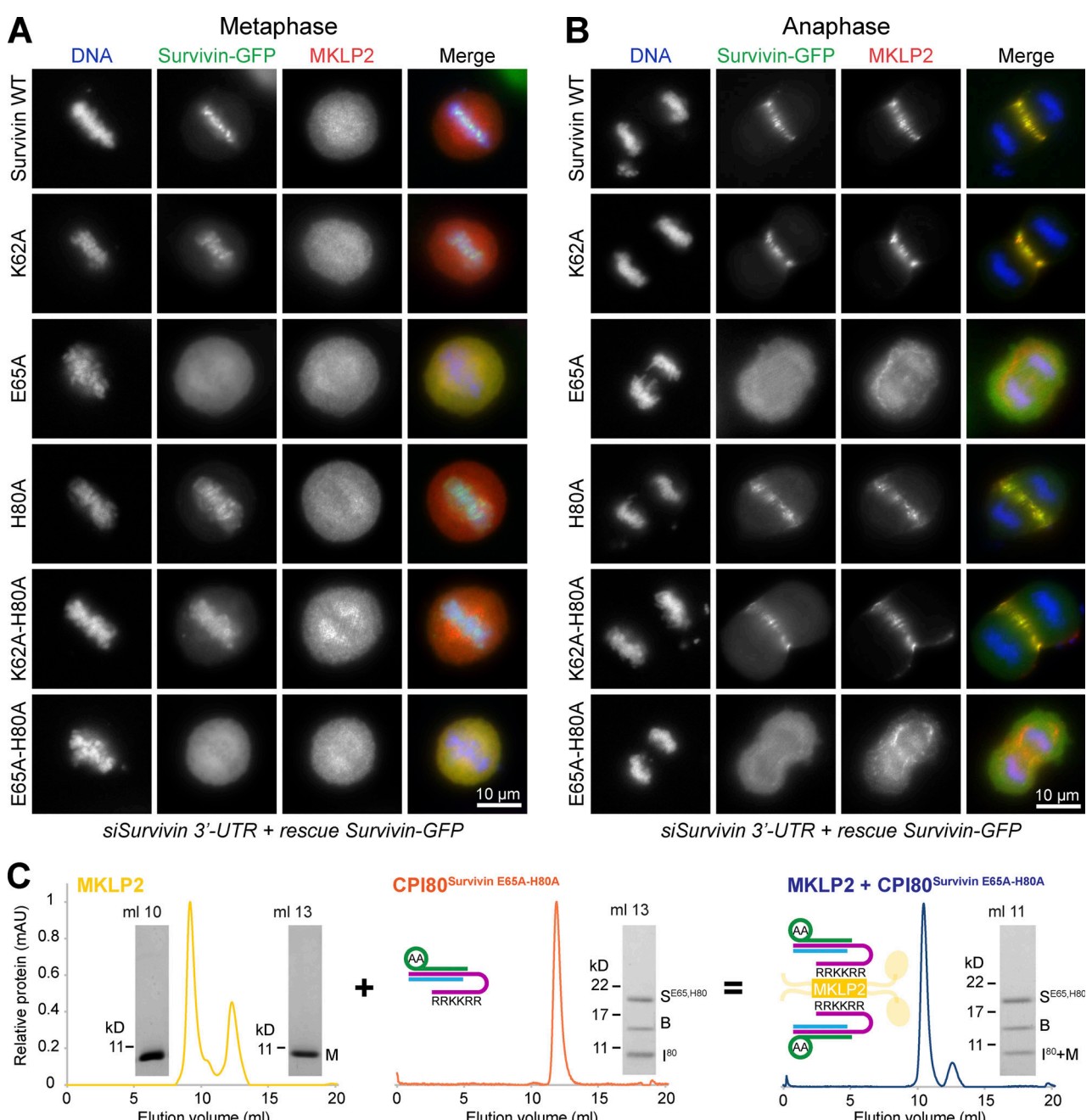

Figure 7. **Survivin-dependent centromere targeting in metaphase is a prerequisite for MKLP2-dependent CPC localization in anaphase. (A and B)** HeLa cells expressing different survivin-GFP constructs (WT, K62A, E65A, H80A, K62A-H80A, and E65A-H80A) were depleted of endogenous survivin using a 3' UTR siRNA and then stained for MKLP2. Representative images of cells in metaphase (A) and anaphase (B) are shown. **(C)** SEC of MKLP2$^{557-668}$ (yellow) with CPI80$^{Survivin E65A-H80A}$ (orange) was performed to test for MKLP2:CPI80$^{Survivin E65A-H80A}$ complexes (blue). In the cartoon, depictions of protein complexes are as follows: survivin is in green, borealin in blue, INCENP in purple, and MKLP2 in yellow. AA denotes the survivin E65A, H80A mutant. mAU, milli-absorbance unit. M, MKLP2; S$^{E65,H80}$, survivin$^{E65,H80}$ mutant; B, borealin; I$^{80}$, INCENP amino acids, 1–80.

with MKLP2. Upon dephosphorylation of INCENP T59 (Hümmer and Mayer, 2009) and MKLP2 at S532, T857, S867, and S878 (Kitagawa et al., 2014), MKLP2 can bind to the CPC, competing for the interaction with DNA. We propose that this complex remains transiently associated with chromatin due to the interaction of survivin with histone H3 (Fig. 10, A and B). Because CPC binding also stimulates the ATPase activity of MKLP2, this transient complex is then extracted from chromatin directly on

to the microtubules, avoiding a soluble intermediate (Fig. 10 C). This ensures MKLP2-dependent movement along microtubules of the anaphase central spindle depends on the presence of the CPC. The behavior of the MKLP2 rigor mutant, which traps Aurora B on chromatin (Fig. S3 A), is also consistent with the idea that ATPase activation and force generation are required for CPC transport. Together, these observations explain the codependency for CPC and MKLP2 localization described previously

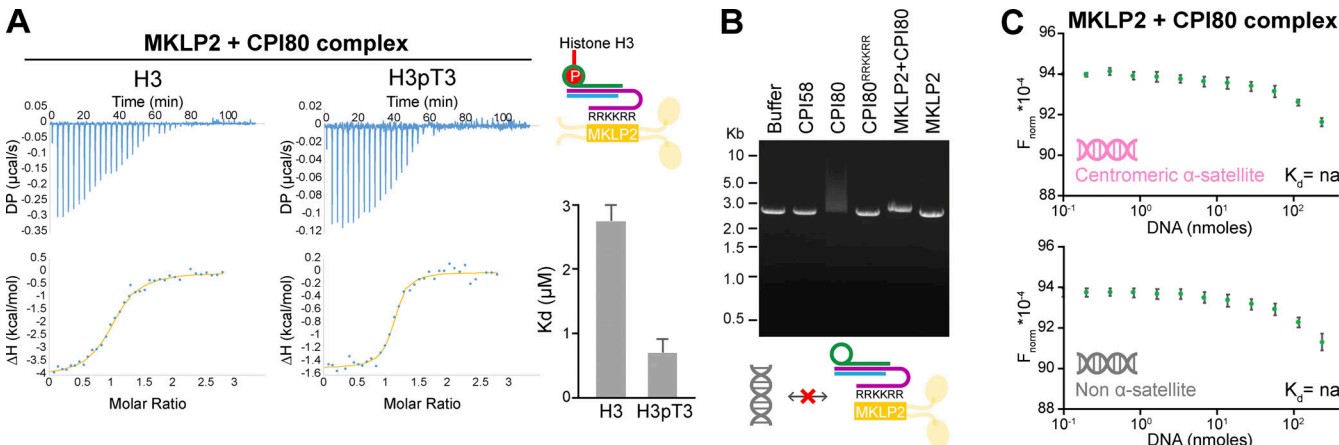

Figure 8. **MKLP2 competes with DNA not H3pT3 for CPC binding. (A)** ITC binding curves for complex formation between MKLP2[557-668]:CPI80 (survivin, INCENP[1-80], borealin[10-109]) and H3 or H3pT3. Histogram bars represent $K_d$ values and error bars represent $K_d$ calculation uncertainty, obtained using MicroCal PEAQ-ITC analysis software. ΔH, enthalpy; DP, differential power. **(B)** EMSA of plasmid DNA incubated with 3.6 µM CPI58, CPI80, CPI80[RRKKRR], MKLP2:CPI80, MKLP2 only, or buffer. **(C)** MST of 0.1–200 µM MKLP2[557-668]:CPI80 (survivin, INCENP[1-80], borealin[10-109]) after incubation with centromeric α-satellite or non–α-satellite DNA duplexes. The graph shows the mean and error bars indicate SEM ($n = 3$), and it was not possible to calculate a $K_d$ values for these conditions. In the cartoon depictions of protein complexes: survivin is in green, borealin in blue, INCENP in purple, MKLP2 in yellow. DNA molecules are colored gray. $F_{norm}$, normalized fluorescence; na, not applicable.

(Hümmer and Mayer, 2009) and provide a simple biochemical switch coupled to the metaphase–anaphase transition.

Why is there such an elaborate mechanism to pick up the CPC from chromatin? We believe that this may relate to the function of Aurora B in cell division. Local gradients of Aurora B at the centromere and then central spindle are crucial for the correction of errors in chromosome alignment and chromosome segregation and for cytokinesis (Afonso et al., 2014; Fuller et al., 2008). The mechanism we propose would ensure there is no freely diffusible pool of active Aurora B complexed to MKLP2 that could interfere with these processes.

There are still unanswered questions relating to this proposed mechanism. One pressing question is the selectivity of the INCENP–DNA interaction. Although both centromeres and neocentromeres are usually assembled on AT-rich DNA, there is little conservation of the underlying DNA sequence (Naughton and Gilbert, 2020). However, it has previously been reported that neocentromeres assembled away from α-satellite regions show less defined CPC targeting and altered Aurora B regulation (Bassett et al., 2010). This results in increased chromosome missegregation and mitotic errors (Naughton and Gilbert, 2020). Thus, while the underlying DNA sequence may not be crucial for centromere and kinetochore formation, it may facilitate recruitment of specific factors such as the CPC, which plays a crucial role in the correction of errors in chromosome alignment and segregation. Our results using AT-rich α-satellite sequences provide some evidence that INCENP RRKKRR-motif–mediated DNA binding has some sequence selectivity, possibly for AT-rich DNA, which may explain these effects. However, our results do not support the view that CPC targeting is strongly sequence dependent. At present, our view is that DNA binding needs to be seen in the context of the CPC–nucleosome interaction and that further studies are required to address this. In addition to the borealin-mediated interaction with nucleosomes (Abad et al., 2019), we

cannot exclude the possibility that the RRKKRR-motif makes additional interactions to other charged surfaces on chromatin, such as the acidic patch formed by histone H2A/H2B on the surface of the nucleosome (Luger et al., 1997). To address all these questions will require structural and functional studies of the CPC bound to DNA-wrapped nucleosomes and detailed comparison of CPC targeting to both centromeres and neocentromeres.

A further question relates to the spatially restricted interaction of MKLP2 with the CPC. Survivin E65A mutants deficient for histone H3 recognition are unable to localize to chromatin and fail to target to the anaphase spindle or microtubule structures, despite being proficient for MKLP2 binding (Fig. 7). This implies that, in cells, MKLP2 binding to the CPC at the onset of anaphase is restricted to sites on chromatin. At present, this dependency for chromatin targeting cannot be fully explained, but it does indicate the presence of further regulation of MKLP2 or the CPC. This is conceivably via the known ubiquitin-dependent regulators of Aurora B function at chromatin (Dobrynin et al., 2011; Krupina et al., 2016; Ramadan et al., 2007). Other work shows that inactive cytoplasmic CPC is chaperoned by the nucleoplasmin family proteins (Hanley et al., 2017), suggesting that this might inhibit binding to other factors such as MKLP2 away from chromatin. The data presented here may indicate an additional possibility, that the N terminus of INCENP contains a regulatory element. Prior studies failed to obtain structures of the CPC with phosphorylated histone H3 peptides, and as we have already noted, that crystal contacts between the N terminus of INCENP and the phospho-binding pocket of survivin occlude the peptide binding site (Fig. S2 A). Interestingly, this sequence at the N terminus of INCENP has two threonine residues that could be potential phospho-acceptors. This suggests it might play a regulatory role in cells, possibly competing for histone H3 binding during the release reaction. One final intriguing possibility involves MKLP2, which was originally identified as a Rab6-binding kinesin at the Golgi apparatus involved in

**Figure 9. Interaction with the INCENP RRKKRR-motif promotes MKLP2 ATPase activity. (A)** Doxycycline-inducible HeLa Flp-In T-REx cells, expressing GFP-INCENP full-length ($f$) WT or RRKKRR $f$ point mutant, were silenced for endogenous INCENP using siINCENP 3' UTR and stained for MKLP2. **(B)** GFP-INCENP enrichment at the central spindle ($f_{central\_spindle}/f_{cytoplasmic}$) is plotted in the graph (WT $n = 30$, RRKKRR $f$ $n = 34$), where the mean with individual data points are marked and error bars indicate SEM. An unpaired $t$ test with Welch's correction and 99% confidence intervals was performed (****, $P < 0.0001$). **(C)** Kinesin ATPase assays were performed using 50 nM full-length MKLP2 or mitotic phosphorylated MKLP2P in the presence or absence of CPI80 or CPI80$^{T59E}$. The mean inorganic phosphate (Pi) released in nanomoles is plotted, with error bars indicating the SD. **(D)** Mean kinesin ATPase activity extracted from the slope at the 15-min time point, relative to the MKLP2-only control, is plotted with error bars indicating the SD for MKLP2 incubated with CPI58, CPI80, CPI80$^{T59E}$, and CPI80$^{RRKKRR}$.

protein transport (Echard et al., 1998) and was then shown to be essential for mitosis (Fontijn et al., 2001; Hill et al., 2000). How these two roles are related remains unclear. Intriguingly, a region overlapping the CPC-binding site contains the binding site for Rab6 (Miserey-Lenkei et al., 2017). This raises the interesting notion that Rab6 competes for, or in some other way modulates, the interaction of MKLP2 with the CPC. Clearly, further work is needed to explore the different possibilities discussed here and fully explain the regulation of MKLP2 and the CPC.

## Materials and methods

### Reagents and antibodies

General laboratory chemicals and reagents were obtained from Sigma-Aldrich and Thermo Fisher Scientific unless specifically indicated. Inhibitor stocks prepared in DMSO were as follows: 5 mM CDK inhibitor flavopiridol (Sigma-Aldrich) and 20 mM MPS1 inhibitor AZ3146 (Tocris Bioscience). Thymidine (100 mM stock; Sigma-Aldrich) and doxycycline (1 mM stock; Sigma-Aldrich) were dissolved in water. Commercially available polyclonal antibodies (pAbs) or mAbs were used for tubulin (mouse mAb DM1A, T6199; Sigma-Aldrich), cyclin B1 (mouse mAb GNS3, 05-373; Millipore), GFP (rabbit pAb, ab290; Abcam),

Histone H3 (rabbit mAb D1H2, 4419; Cell Signaling), Histone H3pT3 (rabbit pAb, 07-424; Millipore), Aurora B (mouse mAb AIM-1, 611083; BD Transduction Laboratories), Survivin (rabbit mAb EP2880Y, ab76424; Abcam), Borealin (mouse mAb A-5, sc-376635; Santa Cruz), Histone H3pS10 (mouse mAb 6G3, 9706; Cell Signaling), and PRC1 pT481 (rabbit mAb EP1514Y, ab62366; Abcam). Human CREST serum was obtained from Antibodies Inc. (15-234-0001). Sheep antibodies produced against full-length Aurora B, MKLP2 (amino acids 63–193; Bastos and Barr, 2010; Neef et al., 2003), and GFP (Bastos and Barr, 2010) have been described previously. Rabbit antibodies were produced against INCENP (IN-box domain), MKLP2 (amino acids 63–193), and PRC1 (Neef et al., 2007). Secondary donkey antibodies against mouse, rabbit, or sheep and labeled with Alexa Fluor 488, Alexa Fluor 555, Alexa Fluor 647, or Cy5, were purchased from Molecular Probes. HRP-conjugated secondary donkey antibodies against mouse, rabbit, or sheep were purchased from Jackson ImmunoResearch Laboratories, Inc. DNA was stained with Hoechst 33258. Affinity-purified primary and HRP-coupled secondary antibodies were used at 1 µg/ml final concentration. For Western blotting, proteins were separated by SDS-PAGE and transferred to nitrocellulose using a Trans-blot Turbo system (Bio-Rad). Protein concentrations were measured by Bradford

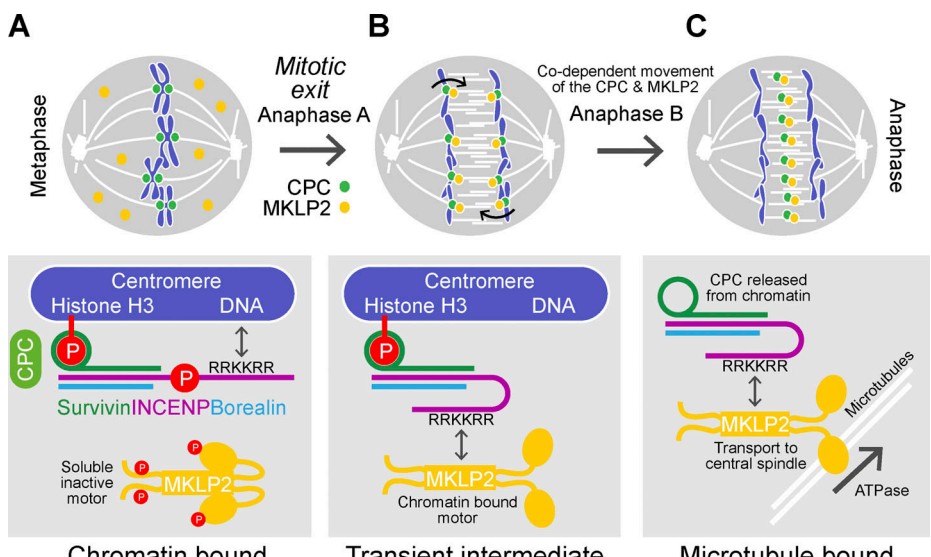

**Figure 10.  A model for MKLP2-dependent Aurora B transport from chromatin to the anaphase central spindle.** CPC (light green) binding to the centromeres is mediated through multiple weak interactions between survivin (dark green) and phosphorylated histone H3, borealin (blue) with nucleosomes, and the INCENP (purple) RRKKRR-motif with DNA. For simplicity, the borealin–nucleosome interaction is not depicted. Phosphorylation of INCENP at T59 and MKLP2 (yellow) at multiple sites prevents interaction of MKLP2 with the CPC. In anaphase, dephosphorylation of INCENP and MKLP2 results in binding of these two proteins, competing for interaction to the DNA. We propose a transient intermediate state where the CPC interacts with MKLP2 but still remains bound to phosphorylated histone H3. CPC-stimulated MKLP2 motor activity then drives movement of MKLP2 with bound CPC away from the chromosome to the anaphase spindle.

assay using Protein Assay Dye Reagent Concentrate (Bio-Rad). All Western blots were revealed using ECL (GE Healthcare). Acrylamide gels staining was performed with InstantBlue (Expedeon).

### Molecular biology
All DNA primers were obtained from Invitrogen. Coding sequences of MKLP2, INCENP, borealin, and survivin were amplified from human testis cDNA (Becton Dickinson) using the *pfu* polymerase (Promega). Mammalian expression constructs for MKLP2, INCENP, and survivin were made in pcDNA5/FRT/TO (Invitrogen) modified to encode EGFP tag (N-terminal tag for MKLP2 and INCENP, C-terminal tag for survivin). Mutagenesis to introduce specific point mutations was performed using the QuickChange method (Agilent Technologies), whereas MKLP2 deletion mutants were obtained by overlapping PCR starting from the WT sequence. siRNA duplexes targeting the 3′ UTR were used to target endogenous MKLP2 (5′-CCACCUAUGUAAUCUCACAUG-3′), INCENP (5′-GGCUUGGCCAGGUGUAUAU-3′), and survivin (5′-GCAGGUUCCUUAUCUGUCA-3′). As a negative control, the luciferase GL2 duplex was used. Bacterial expression constructs for INCENP and MKLP2 were made in the pFAT2-His$_6$-GST plasmid, with a human rhinovirus (HRV) 3C protease cleavage site inserted between GST and the insert. The pETDuet-1 expression vector (Addgene) was used to coexpress survivin and borealin: the former was cloned into the multiple cloning site I with an N-terminal His tag, the latter was cloned into the multiple cloning site II. Recombinant baculovirus carrying His$_6$-tagged full-length MKLP2 cloned into the pFastBac vector was produced using the Baculovirus Expression System (Invitrogen).

### Mammalian and insect cells culture
HeLa and HeLa Flp-In T-REx cells were cultured in DMEM with 1% (vol/vol) GlutaMAX (Life Technologies) containing 10% (vol/vol) bovine calf serum at 37°C and 5% $CO_2$. For plasmid transfection and siRNA transfection, Mirus LT1 (Mirus Bio LLC) and Oligofectamine (Invitrogen), respectively, were used. In both cases, transfection mix was prepared in Opti-MEM. HeLa Flp-In T-REx cell lines with single integrated copies of full-length WT GFP-INCENP point mutants T59E, T59A, RRKRR, and truncation mutants amino acids 1–58, 1–80, 1–80 RRKKRR to hexa-alanine were created using the T-REx doxycycline-inducible Flp-In system and selection with hygromycin (200 µg/ml; Invitrogen). Sf9 insect cells were grown in insect cell growth medium (TC100 containing 10% [vol/vol] bovine calf serum and GlutaMAX) at 27°C and atmospheric $CO_2$.

### Immunofluorescence microscopy of fixed cells
Cells were seeded in six-well plates on no. 1.5 thickness glass coverslips (Menzel-Gläser; Thermo Fisher Scientific). In siRNA rescue assays, either doxycycline induction or plasmid transfection was performed 6 h before silencing. For synchronization, cells were treated for 18 h with 2 mM thymidine and washed three times in PBS and twice with growth medium. Cells were then fixed with 3% (wt/vol) paraformaldehyde in PBS followed by quenching with 50 mM $NH_4Cl$ in PBS and a 5-min cell permeabilization with 0.2% (vol/vol) Triton X-100 in PBS. Antibody dilutions were performed in PBS: affinity-purified antibodies were used at 1 µg/ml, whereas commercial antibodies were used according to manufacturers' instructions. Hoechst 33258 was added to the secondary antibody staining solution at 0.3 µg/ml. Coverslips were mounted in Mowiol 4-88 mounting medium

(EMD Millipore). Samples were imaged using a 60×/1.35 NA oil immersion objective on a BX61 Olympus microscope equipped with filter sets for DAPI, EGFP/Alexa Fluor 488, Alexa Fluor 555, and Alexa Fluor 647 (Chroma Technology Corp.) and a Photometrics Prime camera. Image stacks with spacing of 0.4 µm through the cell volume were taken and then converted in RGB TIFF files using MetaMorph 7.8 imaging software (Molecular Devices). Images were then cropped in Adobe Photoshop and placed into Adobe Illustrator to produce the figures.

### Immunoprecipitation of CPC and MKLP2 complexes

HeLa cells were arrested with 0.1 µM nocodazole for 16 h, and mitotic cells were collected by shake-off and washed three times with PBS and twice with growth medium warmed to 37°C. Metaphase cells were collected after incubation in DMEM at 37°C for 25 min. Anaphase cells were collected at different time points after the addition of AZ3146. At each time point, cells were washed with ice-cold PBS. Cell pellets were suspended in mitotic lysis buffer (20 mM Tris-HCl, pH 7.4, 150 mM NaCl, 1% [vol/vol] Igepal CA-630, 0.1% [wt/vol] sodium deoxycholate, 100 nM okadaic acid, 40 mM β-glycerophosphate, 10 mM NaF, 0.3 mM $Na_3VO_4$, protease inhibitor cocktail [Sigma-Aldrich], and phosphatase inhibitor cocktail [Sigma-Aldrich]). The cell lysate was supplemented with 2 mM $MgCl_2$ before addition of 40 U of Benzonase (Sigma-Aldrich) and then incubated for 30 min on ice, 30 min at room temperature, and 30 min on ice. Proteins were then clarified by centrifugation, and protein complexes were isolated from 3 mg of cell lysate using 3 µg of sheep anti-GFP or anti-MKLP2 antibodies and 20 µl of protein G Sepharose. Isolated complexes were washed once with lysis buffer and three times with wash buffer (20 mM Tris-HCl, pH 7.4, 150 mM NaCl, 40 mM β-glycerophosphate, 10 mM NaF, 0.3 mM Na-vanadate, and 0.1% [vol/vol] Igepal). Samples were then re-suspended in 3× Laemmli buffer for Western blotting.

### Analysis of GFP-MKLP2 from HeLa cells by SEC

GFP-MKLP2 (WT or Δ636-652)–transfected HeLa cells were arrested with 0.1 µM nocodazole for 16 h, and mitotic cells were collected by shake-off. Cell pellets were suspended in mitotic lysis buffer (20 mM Tris-HCl, pH 7.4, 200 mM NaCl, 1% [vol/vol] Igepal CA-630, 0.1% [wt/vol] sodium deoxycholate, 100 nM okadaic acid, 40 mM β-glycerophosphate, 10 mM NaF, 0.3 mM $Na_3VO_4$, protease inhibitor cocktail [Sigma-Aldrich], and phosphatase inhibitor cocktail [Sigma-Aldrich]). Cell lysates were then supplemented with 2 mM $MgCl_2$ before addition of 40 U of Benzonase and incubated for 30 min on ice, for 30 min at room temperature, and for 30 min on ice. Proteins were then clarified by centrifugation, and 1 mg of cell lysate was analyzed by SEC using a Superose 6 10/300 column (GE Healthcare). All chromatography steps were performed at 4°C using a Prime Plus System (GE Healthcare). Eluted fractions were then analyzed by Western blot.

### Bacteria cell culture and purification of CPC complexes and MKLP2

BL21-CodonPlus (DE3)-RIL *Escherichia coli* competent cells (Stratagene) were cultured in Luria–Bertani media in the presence of 100 µg/ml ampicillin and 34 µg/ml chloramphenicol. DNA transformation was performed by heat shock, and protein expression was induced overnight at 18°C after the addition of 0.1 mM IPTG. Cells were harvested by centrifugation at 4°C and lysed in wash buffer (500 mM NaCl, 20 mM Tris-HCl, pH 8.0, and 1 mM DTT), supplemented with 0.5 mg/ml lysozyme, 5 U/ml DNase (Roche), and 1 tablet of Complete Ultra Mini EDTA-free Protease Inhibitor Cocktail (Roche) per liter of culture. For CPC purification, INCENP and survivin–borealin pellets were lysed together. After sonication and centrifugation, the supernatant was incubated for 2 h with Glutathione Sepharose 4B (GE Healthcare) and then loaded into a column. After washing with wash buffer, protein complexes were eluted with 20 mM reduced glutathione. Tags were cleaved away from protein complex during overnight dialysis at 4°C against wash buffer, in the presence of the GST-tagged HRV 3C protease (1:100, wt/wt). The CPC was then purified by SEC in wash buffer supplemented with 10% glycerol using a HiLoad 16/600 Superdex 75-pg column (GE Healthcare) followed by GSTrap 4B columns (GE Healthcare). If necessary, the complex was further purified by anion exchange using a Mono Q 5/50 GL column (GE Healthcare), with a gradient of NaCl. MKLP2 purification was similarly performed, but using a HiLoad 16/600 Superdex 200-pg column (GE Healthcare). MKLP2:CPC complexes were obtained after incubation of CPC variants and MKLP2[557-668] in a 1:1 molar ratio for 30 min in ice in binding buffer (100 mM NaCl, 20 mM Tris-HCl, pH 8.0, 1mM DTT, and 10% glycerol), followed by SEC using a HiLoad 16/600 Superdex 200-pg column. MKLP2- and CPC-binding assays were performed on an analytical Superdex 200 increase 10/300 column (GE Healthcare). All chromatography steps were performed at 4°C using an EttanLC System (GE Healthcare).

### Crystallization and data collection

CPI[NT] equivalent to previous work (survivin, INCENP[1-58], borealin[10-109]; Jeyaprakash et al., 2007), CPI58 (survivin, INCENP[7-58], borealin[10-76]), CPI80 (survivin, INCENP[7-80], borealin[10-76]), and MKLP2[596-668] were dialyzed against 50 mM MES, pH 6.8, and 2 mM DTT and concentrated up to 10 mg/ml. For peptide binding, H3pT3 (ART[P]KQTARKSTG) was incubated with the CPC in a 2:1 molar ratio. CPI[NT] crystals were obtained in 50 mM MES, pH 6.0, 5% (wt/vol) polyethylene glycol (PEG) 3350. CPI58:H3pT3 crystals were obtained in 0.2 M lithium sulfate, 0.1 M MES, pH 6.5, and 20% (wt/vol) PEG 400. CPI80:H3pT3 crystals were obtained in 0.15 M ammonium sulfate, 0.1 M Hepes-NaOH, pH 7.0, and 20% (wt/vol) PEG 400. MKLP2[596-668] crystals were obtained in 0.1 M sodium citrate, pH 5.6, 20% (vol/vol) 2-propanol, and 20% (wt/vol) PEG 400. When needed, 20% (vol/vol) PEG 400 or 2-methyl-2,4-pentanediol, prepared using mother liquor, was used as cryoprotectant. CPI[NT] diffracted to 3.5-Å resolution at the P14 beamline operated by EMBL Hamburg at the PETRA III synchrotron (DESY, Hamburg, Germany). CPI58:H3pT3 and MKlp2[596-668] crystals diffracted to 1.81 and 1.43 Å, respectively, at the I24 beamline at Diamond Light Source synchrotron (Didcot, UK). CPI80:H3pT3 crystals diffracted to 2.55-Å resolution at the ID23-1 beamline at the European Synchrotron Radiation Facility (Grenoble, France) synchrotron.

## Crystal structure solution and refinement

CPC data were processed and integrated using DIALS (Winter et al., 2018), and crystal structures were determined by molecular replacement with PHASER (McCoy et al., 2007) using PDB accession no. 2QFA (Jeyaprakash et al., 2007) as a model. MKLP2 data were processed and integrated using autoPROC (Vonrhein et al., 2011) and STARANISO (Tickle et al., 2018; Global Phasing Ltd). The structure was solved using PHASER (McCoy et al., 2007) and PDB accession no. 5LEF (Miserey-Lenkei et al., 2017) as a model. Iterative rounds of model building and refinement were performed with COOT (Emsley and Cowtan, 2004) and PHENIX (Adams et al., 2010), respectively. Data collection and refinement statistics are presented in Table S1. Interactions were calculated using CCP4-PISA (Krissinel and Henrick, 2007). All structure figures were generated with PyMOL (https://pymol.org/2/).

## Accession numbers

Coordinates and structure factors have been deposited within the PDB under the following accession codes: CPI$^{NT}$, 6YIE; CPI58: H3pT3, 6YIF; CPI80:H3pT3, 6YIH; and MKLP2$^{596-668}$, 6YIP.

## ITC of the CPC–peptide interaction

All peptides were synthesized by Peptide Protein Research Ltd: H3 (1–12: ARTKQTARKSTG), H3pT3 (1–12: ART$^P$KQTARKSTG), H2A (119–129: KKTESHHKAKG), and H2ApT121 (119–129: KKT$^P$ESHHKAKG). Lyophilized peptides were dissolved against ITC buffer (100 mM NaCl, 20 mM Tris-HCl, pH 7.2, and 1 mM DTT). Protein samples were dialyzed overnight at 4°C against the same ITC buffer. All the binding experiments were performed on a MicroCal PEAQ-ITC instrument (Malvern Panalytical) at 25°C. Protein concentration was 50 µM, whereas peptide concentration was 400 µM. First titration was performed using 0.5 µl of peptide, followed by 19 2-µl injections applied 180 s apart. Data were fitted using MicroCal PEAQ-ITC software with a 1:1 binding model.

## EMSAs for DNA binding

Assays were performed with BamHI-linearized pBluescript plasmid (0.15 pmol), 3.6 µM the different CPC complexes, and control proteins in 20 µl of 100 mM NaCl, 20 mM Tris-HCl, pH 7.4, and 1 mM DTT for 2 h on ice. Reactions were analyzed on a 0.8% (wt/vol) agarose gel in the presence of 0.8 µg/ml ethidium bromide. For CPI80 titration, the concentration range indicated in Fig. 3 was used. Images of the ethidium bromide–stained gel were taken using a UVP BioDoc-It² Imager (Analytik Jena).

## Measurement of DNA binding using MST

MST experiments were performed on a Monolith NT.115 instrument using a blue filter (NanoTemper Technologies) and Alexa 488–labeled DNA for non–α-satellite DNA (488-forward: 5′-ATTCCATGGCACCGTCAAGGCTGAGAACGGGAAGCTTGTC-3′; reverse: 5′-GACAAGCTTCCCGTTCTCAGCCTTGACGGTGCCAT GGAAT-3′) and a centromeric human α-satellite DNA consensus (Vissel and Choo, 1987; 488-forward: 5′-ATTCAACTCACAGAG TTGAACCTTCCTTTTCATAGAGCAG-3′; reverse: 5′-CTGCTCTAT GAAAAGGAAGGTTCAACTCTGTGAGTTGAAT-3′). The α-satellite or non–α-satellite DNA duplexes were annealed in MST buffer (100 mM NaCl, 20 mM Tris-HCl, pH 7.4, and 1 mM DTT). For

DNA-binding assays, a dilution series of proteins from 200 to 0.1 µM and 0.04 µM annealed α-satellite or non–α-satellite DNA duplexes were incubated in MST buffer for 2 h in ice before being loaded into Monolith NT.115 standard treated capillaries (Nano-Temper Technologies). Measurements were performed with power settings of 80% for MST and 50% for the light-emitting diode. Data analysis and $K_d$ calculation were performed using the NanoTemper analysis software.

## SEC-MALS

Samples, previously purified by SEC, were diluted to 1 mg/ml and analyzed by SEC-MALS. Experiments were performed at room temperature during SEC on an analytical Superose 6 10/300 GL column, equilibrated with 100 mM NaCl, 20 mM Tris-HCl, pH 8.0, 1 mM DTT, and 10% (vol/vol) glycerol. Elution was monitored via online static light-scattering (DAWN HELEOS 8+; Wyatt Technology), differential refractive index (Optilab T-rEX; Wyatt Technology) and UV (SPD-20A; Shimadzu) detectors. Data were analyzed using the ASTRA software package (Wyatt Technology).

## Expression and purification of MKLP2 from insect cells

Recombinant baculoviruses encoding His$_6$-tagged MKLP2 used to infect $4 \times 10^7$ insect cells in 20 × 15-cm dishes with a multiplicity of infection of 10. To infect cells, the medium was removed, and the virus was added in 3 ml of insect growth medium per dish. Dishes were gently rocked for 1 h; every 10 min the dishes were rotated by 90°. After 1 h, 17 ml of insect cell growth medium was added, and the cells were left for 60 h before harvesting by centrifugation at 200 $g$. To obtain M-phase, CDK-phosphorylated MKLP2, the cells were treated for 3 h with 100 nM okadaic acid before harvesting. Cell pellets were washed once in ice-cold PBS and then lysed in 10 ml of IMAC20 (20 mM Tris-HCl, pH 8.0, 300 mM NaCl, 20 mM imidazole, and 0.2% [vol/vol] Triton X-100), and protease inhibitor cocktail (Sigma-Aldrich) for 20 min on ice. A cell lysate was prepared by centrifugation at 100,000 $g$ in a TLA100.3 rotor (Beckman Coulter) and then loaded onto a 1-ml HisTrap FF column (GE Healthcare) at 0.5 ml/min. The column was then washed with 30 ml of IMAC20 and eluted with a 20-ml linear gradient from 20 to 200 mM imidazole in IMAC20, collecting 1-ml fractions. Peak fractions judged by SDS-PAGE and absorbance at 280 nm were concentrated using Ultracel-10K centrifugal filters (EMD Millipore) according to the manufacturer's instructions to a final volume of ~1 ml. Samples were then buffer exchanged using 5 ml of Zeba Desalt Spin columns (Perbio) into 20 mM Tris-HCl, pH 8.0, 300 mM NaCl, and 1 mM DTT. Protein samples were snap-frozen in 15-µl aliquots and stored at –80°C for further use.

## MKLP2 kinesin ATPase assays

A commercial enzyme-linked inorganic phosphate assay (BK060; Cytoskeleton, Inc.) was used to measure MKLP2 ATPase activity. Taxol-stabilized microtubules were produced by incubating 45 µl (7.4 mg/ml) of porcine tubulin, 4.5 µl of cushion buffer (80 mM Pipes-KOH, pH 6.9, 2 mM MgCl$_2$, 0.5 mM EGTA, and 60% [vol/vol] glycerol) and 0.5 µl 100 mM GTP at 30°C for 20 min. To stabilize microtubules, 8 µl of 2 mM paclitaxel in 750 µl of tubulin buffer (80 mM Pipes-KOH, pH 6.9, 2 mM MgCl$_2$, and 0.5 mM EGTA) was

added to the microtubule polymerization reaction. This 3.75 µM tubulin stock mix was stored at room temperature until further use. A microtubule premix was created by mixing 1 ml of reaction buffer (15 mM Pipes-KOH, pH 7.0, and 5 mM MgCl₂), 10 µl of 2 mM paclitaxel, 235 µl of Taxol-stabilized microtubules, 240 µl of 1 mM 2-amino-6-mercapto-7-methylpurine riboside, and 12 µl of 0.1 U/µl purine nucleoside phosphorylase. Reactions were set up in 96-well plates by mixing the proteins of interest in a total volume of 7.5 µl. Reactions were set up in 96-well plates by mixing the protein of interest in 150 µl of microtubule reaction mix. To start the assay, 10 µl of 10 mM ATP was added to each well. This was then rapidly transferred to a 37°C plate reader (Tristar LB 941; Berthold Technologies) set to read absorbance at 360 nm. Readings were acquired every 30 s for 1 h. An inorganic phosphate standard curve was created in the same assay buffer and used to convert absorbance to nanomoles of hydrolyzed ATP.

### Quantification of image data and statistical analysis

INCENP localization either on centromeres or the central spindle was calculated as a ratio between the fluorescence measured, respectively, on chromatin ($f_{chromatin}$) or at the central spindle ($f_{central\_spindle}$), divided by the fluorescence measured into the cytoplasm ($f_{cytoplasmic}$). H3pS10 relative intensity ($f_{pS10}/f_{GFP-INCENP}$) was calculated normalizing the fluorescence value for H3pS10 intensity ($f_{pS10}$) to the respective INCENP fluorescence value detected on centromeres ($f_{GFP-INCENP}$) of the same cell. All fluorescence values were corrected by subtracting background intensity.

Statistical analysis for immunofluorescence data was performed using the unpaired $t$ test with Welch's correction, as indicated in the figure legends. For the $t$ test, data distribution was assumed to be normal, but this was not formally tested. Statistical significance was determined using Prism version 5 (GraphPad Software).

MST data were analyzed by two-way ANOVA, considering dilution factor and DNA sequence (centromeric α-satellite or non–α-satellite) as independent variables, using StataMP 14.0 (Stata Corp.). A significance level of $P < 0.05$ was accepted as statistically significant.

### Online supplemental material

Fig. S1 extends the cell line characterization and CPC localization data presented in Fig. 1. The structure of CPI$^{NT}$ showing crystal contacts involving the INCENP N terminus and further structural details of phospho-histone binding to CPI58 and CPI80 are presented in Fig. S2 and extend Fig. 2. Refinement statistics for the structures are presented in Table S1. Original data traces for the MST experiments shown in Fig. 3 are provided in Table S2. Fig. S3 shows the mapping experiments used to identify the CPC-binding domain in MKLP2 in support of Fig. 5. A structure of the CPC-binding region of MKLP2 and additional control data for Fig. 6 are shown in Fig. S4. Additional analysis of survivin- and INCENP-expressing cell lines is provided in Fig. S5.

## Acknowledgments

We would like to thank Dr. Ed Lowe and Diamond Light Source, Harwell, UK for beamtime (proposal mx18069) and the staff of beamline I24 for assistance with crystal testing and data collection. Additional crystal teting and data collection were performed on beamline ID23-1 at the European Synchrotron Radiation Facility, Grenoble, France and PETRA III P14 at the Deutsche Elektronen Synchrotron, Hamburg, Germany. We also thank Dr. Ekaterina P. Lamber for assistance with crystallographic data collection and Dr. David Staunton for assistance with ITC, MST, and SEC-MALS. We thank the Barr and Gruneberg groups for their advice and encouragement during the work.

M. Serena and R.N. Bastos were supported by a Cancer Research Program award (C20079/A15940) to F.A. Barr.

The authors declare no competing financial interests.

Author contributions: Conceptualization: F.A. Barr. Investigation: M. Serena and R. Nunes Bastos. Data analysis: M. Serena, R. Nunes Bastos, P.R. Elliott, and F.A. Barr. Funding acquisition: F.A. Barr. Supervision: F.A. Barr and P.R. Elliot. Writing–original draft: F.A. Barr and M. Serena. Writing–review and editing: M. Serena, R. Nunes Bastos, P.R. Elliott, and F.A. Barr.

Submitted: 10 October 2019

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

# Supplemental material

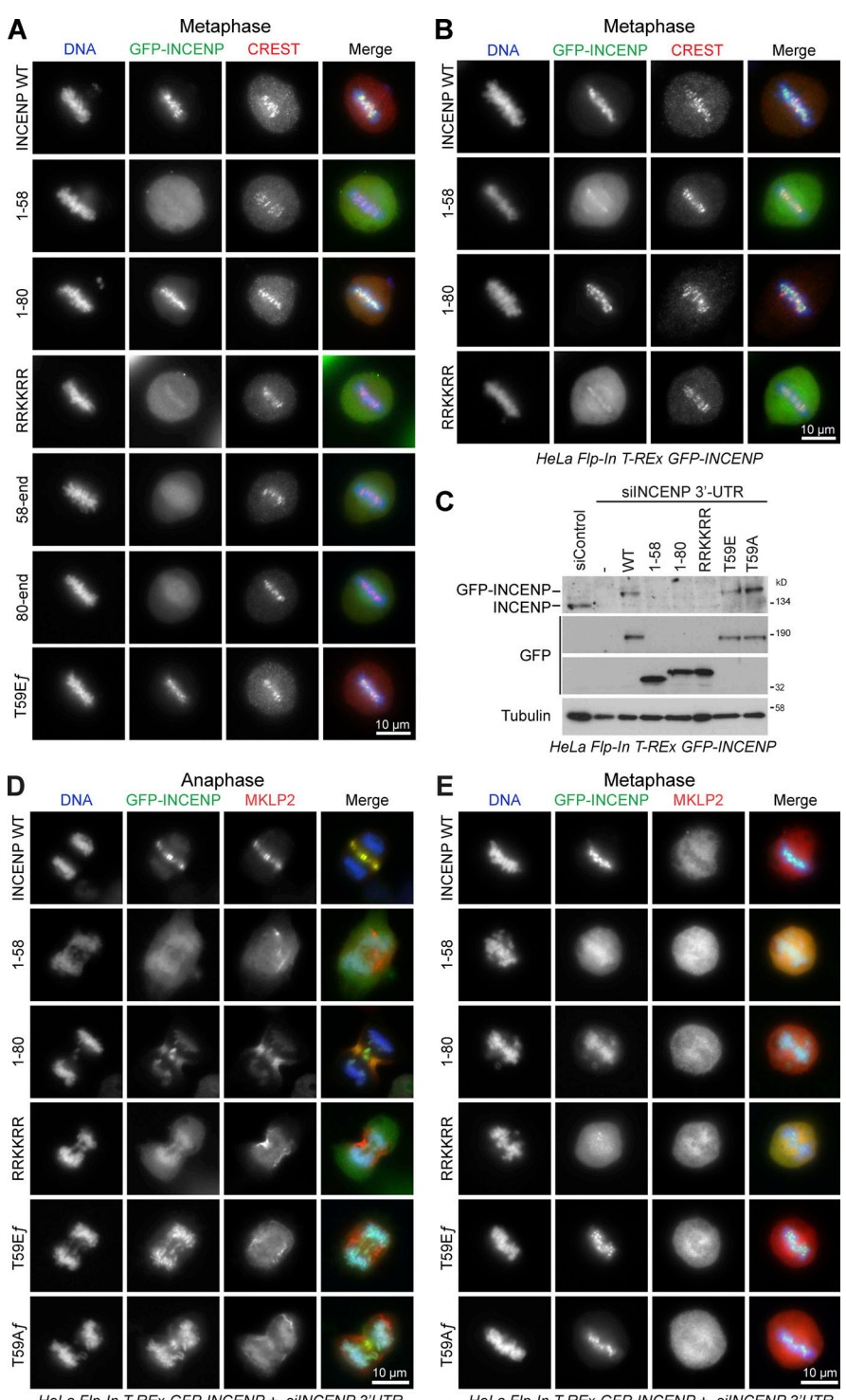

Figure S1.  **INCENP and stable cell lines. (A)** HeLa cells, transiently expressing different GFP-INCENP constructs (full-length [*f*] WT; the T59E *f* point mutant; or truncations 1–58, 1–80, 1–80 RRKKRR [1–80 RRKKRR to hexa-alanine mutant], 58-end, and 80-end) were stained for the centromere marker CREST. Representative images of thymidine-treated cells in metaphase are shown. For INCENP[1-58], 62.5% of cells showed only cytoplasmic targeting and 16.7% showed weak centromere targeting. For INCENP[1-80] RRKKRR, 57.1% of cells showed only cytoplasmic targeting and 21.4% showed weak centromere targeting. **(B)** HeLa Flp-In T-Rex cells, expressing doxycycline-inducible GFP-INCENP constructs (full-length WT or truncations 1–58, 1–80, and RRKKRR) were induced and then stained for the centromere marker CREST. Metaphase cells are shown. **(C)** HeLa Flp-In T-Rex cells, without a transgene (vertical dash) or expressing doxycycline-inducible GFP-INCENP constructs (full-length WT, T59E *f* or T59A *f* point mutants, or truncations 1–58, 1–80, and RRKKRR) were depleted of endogenous INCENP using a 39 UTR siRNA and then Western blotted after doxycycline induction. Tubulin was used as a loading control. The INCENP antibody recognizes an epitope in the C terminus and does not detect the N-terminal fragments. **(D and E)** The induced cells were stained for MKLP2, and representative images of cells in either anaphase (D) or metaphase (E) are shown. Note that N-terminal INCENP fragments unable to bind Aurora B failed to support full chromosome alignment and exhibited lagging chromatin in anaphase.

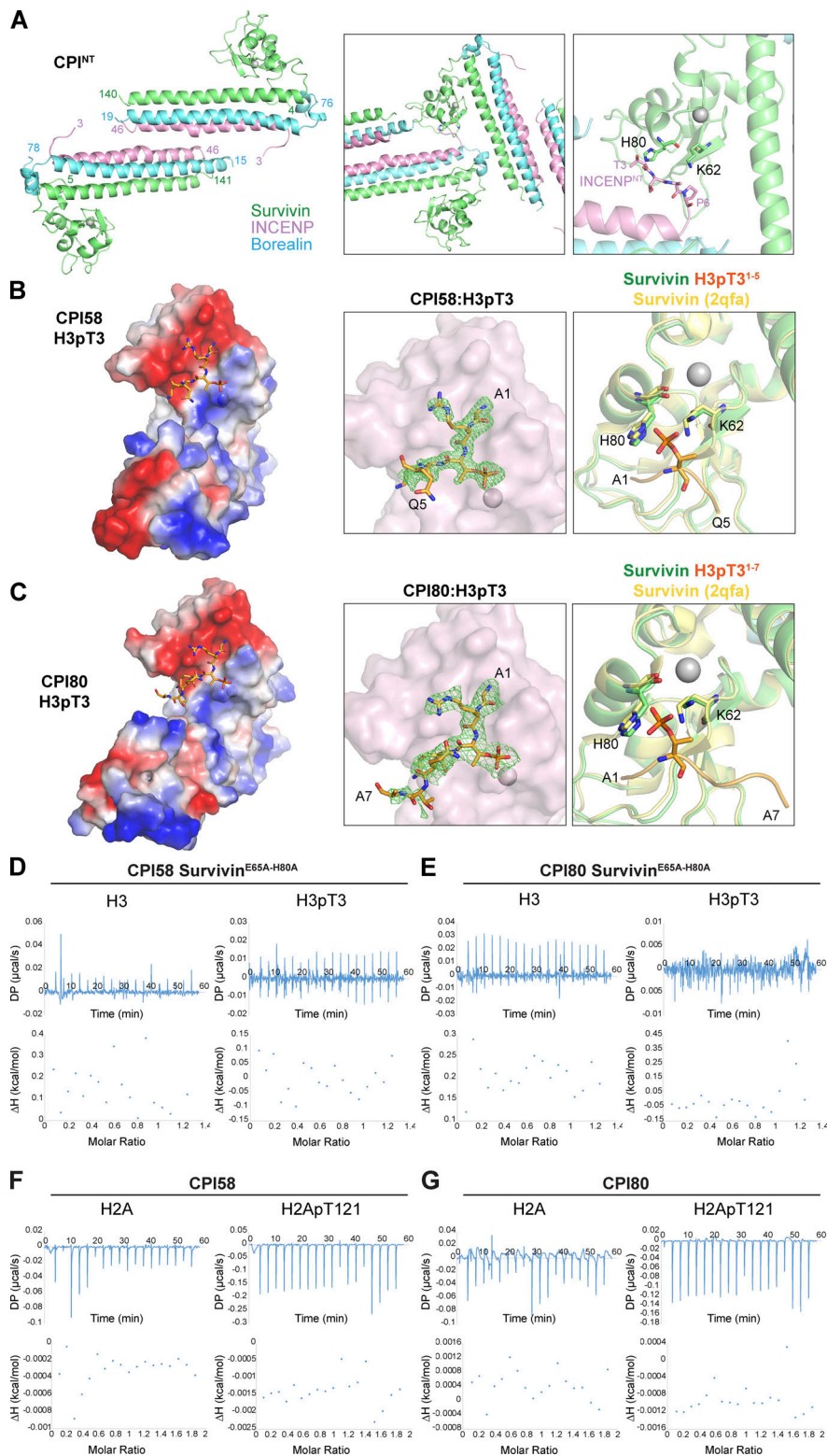

Figure S2. **Structure of CPI58 and CPI80 peptide complexes. (A)** CPI[NT] (survivin in green, borealin in light blue, and INCENP in pink) crystallized as a dimer, inhibiting H3pT3 peptide binding. INCENP N-terminal region of the symmetry-related molecule is interacting with survivin, occupying the same region where the peptide should have bound. Survivin H80 and K62 are shown. **(B and C)** Electrostatic potential maps of the CPI58 (B) and CPI80 (C) surfaces shows the negatively charged (red) peptide-binding groove and the positively charged (blue) cleft where the T3 phosphate sits. The green density map indicates the presence of H3pT3 peptide bound to survivin of both CPI58 (B) and CPI80 (C). Superimposition of CPI58 (B) and CPI80 (C) with PDB accession no. 2QFA (yellow) Jeyaprakash et al. (2007) highlights the different orientations of the side chain of Lys62 of survivin (green) in the presence of H3pT3, confirming its role in phosphate recognition, as previously observed for survivin (Du et al., 2012). **(D–G)** ITC data plots of survivin[E65A-H80A] CPI58 (D) and CPI80 (E) mutants in the presence of H3 or H3pT3. ITC data plots of CPI58 (F) and CPI80 (G) in the presence of H2A or H2ApT121. ΔH, enthalpy; DP, differential power.

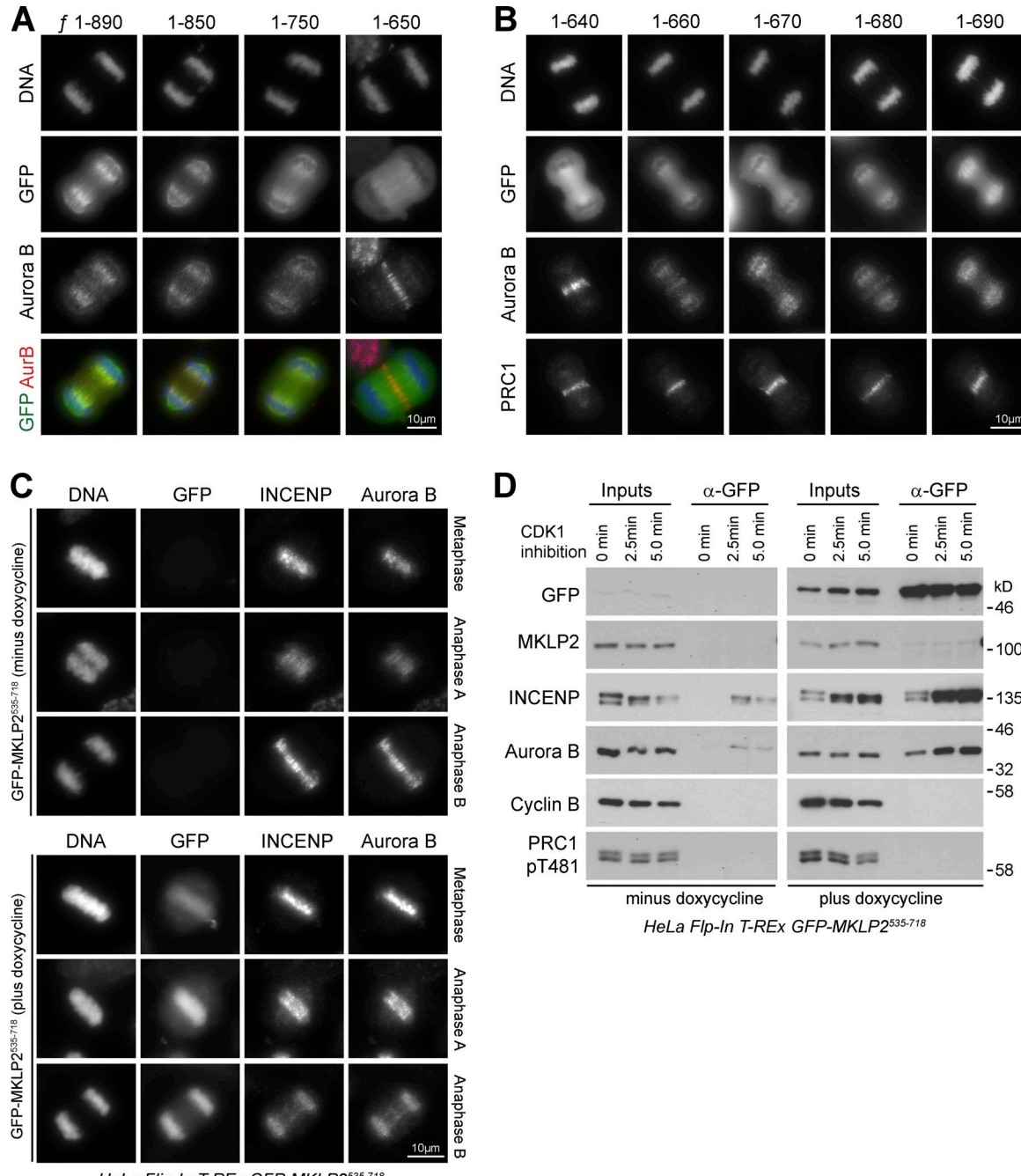

Figure S3. **Mapping the CPC-binding domain using an MKLP2 rigor mutant trap assay. (A)** HeLa cells transfected with full-length (amino acids 1–890) GFP-MKLP2 E413A ATPase defective rigor mutant and C-terminal truncations to positions 850, 750, and 650 were fixed and then stained for Aurora B (red) and DNA (blue). GFP fluorescence for MKLP2 (green) was visualized directly. **(B)** Comparison of amino acid truncations from MKLP2 position 640 to 690. Cells were stained for DNA, Aurora B, and PRC1 to mark the central spindle. GFP fluorescence for MKLP2 was visualized directly. Representative cells in anaphase are shown. **(C)** HeLa T-REx GFP-MKLP2[535-718] cells were left uninduced or induced with doxycycline for 18 h and then fixed and stained for INCENP and Aurora B. Representative cells in metaphase, anaphase A, and anaphase B are shown. **(D)** HeLa T-REx GFP-MKLP2[535-718] cells were left uninduced or induced with doxycycline for 18 h and then arrested in mitosis with nocodazole for 18 h. Cells were forced into anaphase using 5 µM flavopiridol to rapidly inhibit CDK1, and samples were collected at the times indicated. MKLP2 complexes were isolated by GFP Iimmunoprecipitation and blotted for the proteins shown in the figure.

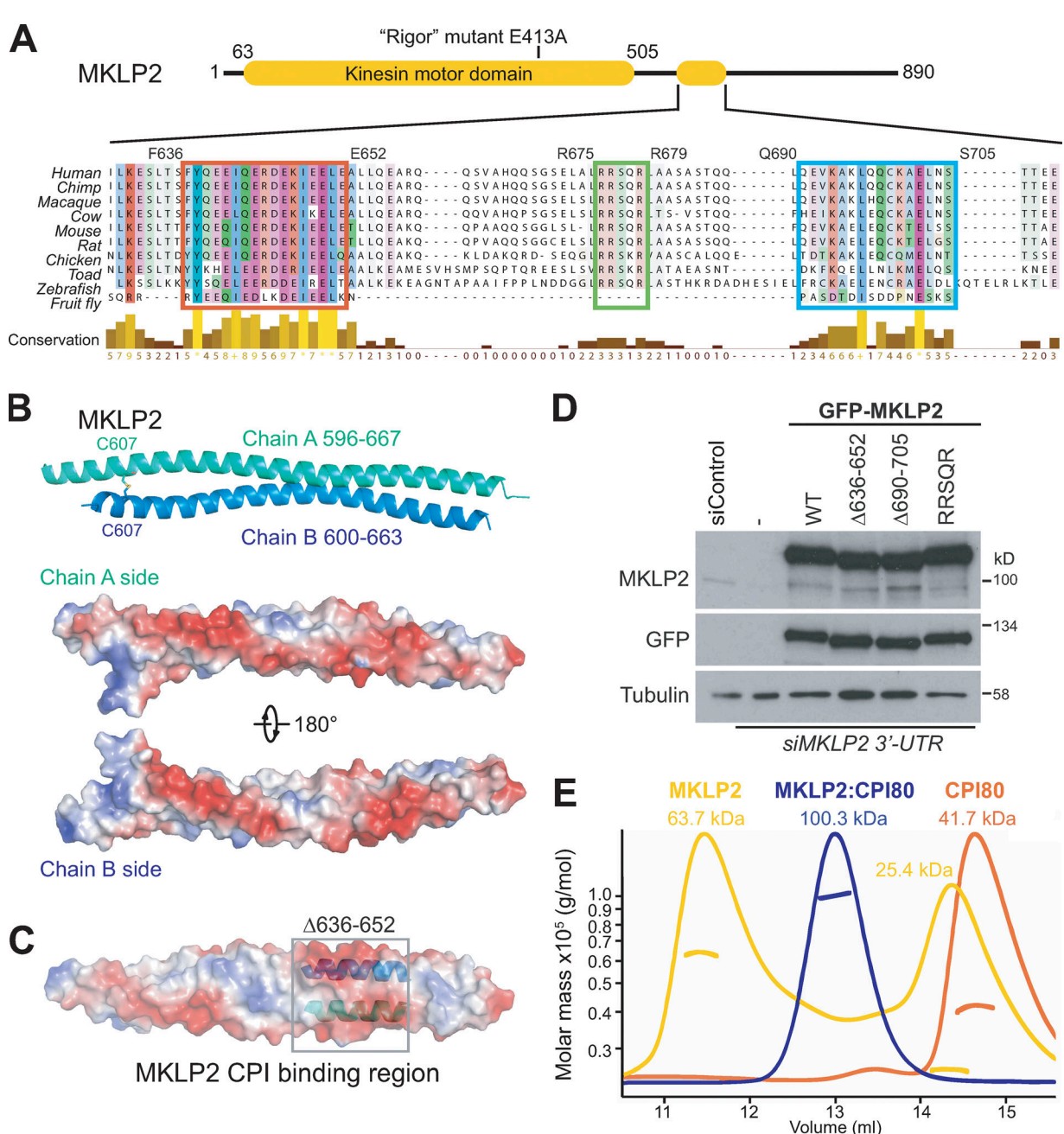

Figure S4. **Structure of the MKLP2 CPC-binding region. (A)** A ClustalX alignment of MKLP2 sequences from different species highlighting different conserved features. **(B)** Crystal structure of dimeric MKLP2$^{596-668}$ at 1.43 Å shows the disulfide bond occurring between C607 of chains A and B. Electrostatic potential on the two sides of MKLP2 indicates a highly negatively charged region (red). **(C)** Electrostatic potential on the surface of MKLP2 fragment 596–668 shows the negatively charged (red) 636–652 region, involved in CPC interaction. **(D)** Western blot of siControl and siMKLP2 cells confirmed depletion of MKLP2 in untransfected control cells (vertical dash) and following rescue with GFP-MKLP2 (WT and mutants). Tubulin was used as a loading control. siControl, nontargeting siRNA. **(E)** SEC-MALS–calculated weight-average molar masses for MKLP2$^{557-668}$ (yellow), CPI80 (orange), and MKLP2:CPI80 (blue) are plotted versus the elution volume. Data were analyzed using the ASTRA software package.

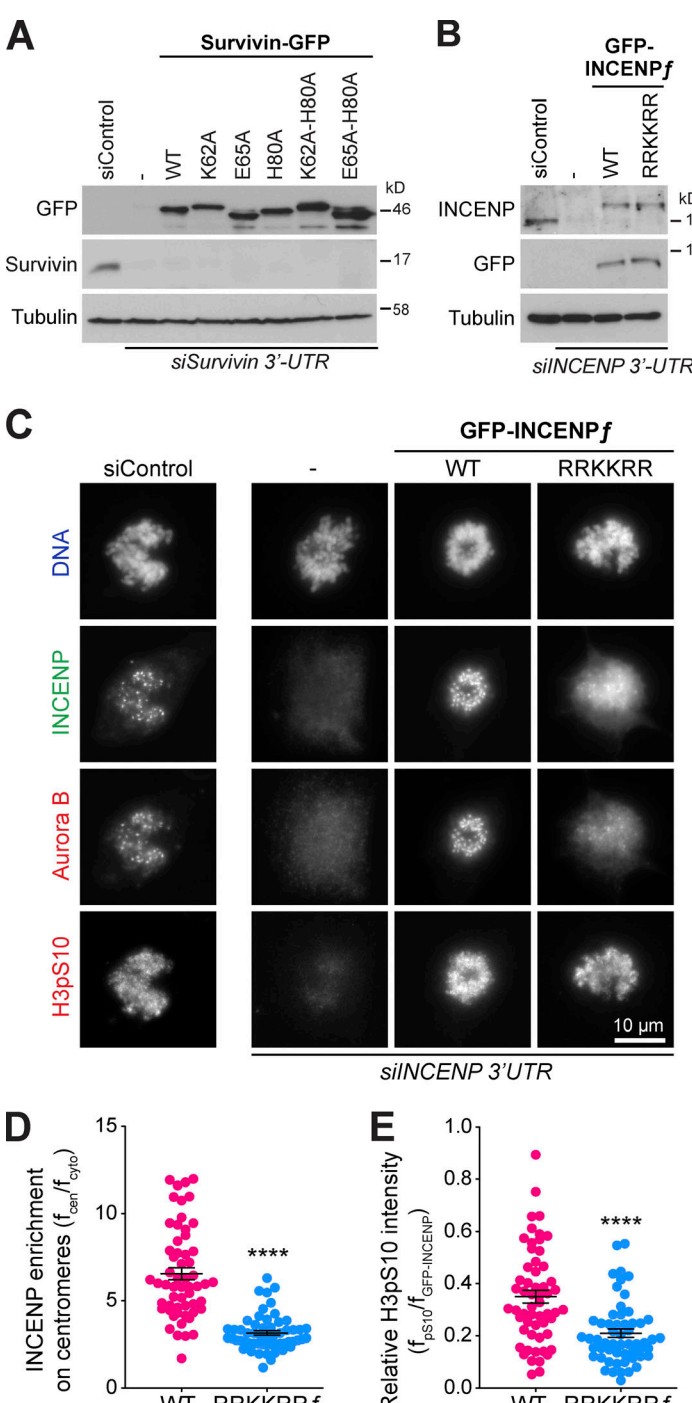

Figure S5. **Survivin mediates centromeric localization of CPC through recognition of Histone H3. (A)** HeLa cells left untransfected (vertical dash) or expressing different survivin-GFP constructs (WT, K62A, E65A, H80A, K62A-H80A, and E65A-H80A) were depleted of endogenous survivin using a 39 UTR siRNA and then Western blotted. Tubulin was used as a loading control. **(B)** HeLa Flp-In T-REx cells, without atransgene (vertical dash) or expressing doxycycline-inducible full-length GFP-INCENP or the RRKKRR-motif mutant, were depleted of endogenous INCENP using a 39 UTR siRNA and then Western blotted following doxycycline induction. Tubulin was used as a loading control . **(C)** Untransfected (horizontal dash) and GFP-INCENP expressing cells were stained for Aurora B and histone H3pS10. Representative images of prometaphase cells are shown. **(D and E)** GFP-INCENP enrichment ($f_{centromere}/f_{cytoplasmic}$) at centromeres (D) and the relative level of histone H3pS10 phosphorylation (E) metaphase chromatin ($f_{pS10}/f_{GFP-INCENP}$) are plotted in the graphs (WT $n$ = 57, RRKKRR $f$ $n$ = 56), where the mean with individual data points is marked and error bars indicate SEM. An unpaired $t$ test with Welch's correction and 99% confidence intervals was performed (****, $P < 0.0001$).

**Tables S1 and S2 are provided online. Table S1 lists data collection and refinement statistics for structures. Table S2 shows original data traces for the MST experiments.**

