## [Peer Review File · The Journal of Cell Biology]

Molecular basis of MKLP2-dependent Aurora B transport from chromatin to the anaphase central spindle

Michela Serena, Ricardo Nunes Bastos, Paul Elliott, and Francis Barr

Corresponding Author(s): Francis Barr, University of Oxford

Review Timeline:

Submission Date:	2019-10-10
Editorial Decision:	2019-11-07
Revision Received:	2020-02-10
Editorial Decision:	2020-03-13
Revision Received:	2020-04-02
Accepted:	2020-04-08

Monitoring Editor: Yixian Zheng

Scientific Editor: Marie Anne O'Donnell

Transaction Report:

DOI: <https://doi.org/10.1083/jcb.201910059>

November 7, 2019

Re: JCB manuscript #201910059

Prof. Francis A Barr
University of Oxford
Department of Biochemistry
South Parks Road
Oxford OX1 3QU
United Kingdom

Dear Prof. Barr,

Thank you for submitting your manuscript entitled "Molecular basis of MKLP2-dependent Aurora B transport from chromatin to the anaphase central spindle". The manuscript was assessed by expert reviewers, whose comments are appended to this letter. We invite you to submit a revision if you can address the reviewers' key concerns, as outlined here.

You will see that the reviewers found your studies exploring how the CPC targets to centromeres from prophase to metaphase and is released to the central spindle in anaphase interesting and generally of high quality. However, they were not yet fully convinced by the data: they were critical of the reliance on transient transfection/overexpression and of the lack of quantifications (Rev#1, point #1; Rev#3 point #2). Rev#1 did not find the significance of the reported DNA-binding capacity of the RRKKRR motif compelling (point #2) -- see also Rev#2's suggestion that you tone down the claim that INCENP promotes binding to alpha-satellite DNA (point #2) and Rev#3's criticisms of the small effects as well (point #3). Rev#2 suggested examining whether T59 phosphorylation affects DNA binding of INCENP with DNA (point #1) and Rev#3 had a few more questions about the model (points #4-5). Rev#1 asked that you confirm that the deletion of MKLP2 636-652 does not affect MKLP2 dimerization and folding (point #5). The referees all brought up the need for additional discussion of past work (Rev#1 minor points and #1, Rev#2 points #3, #4, #5; Rev#3 point #1).

We have editorially considered these issues and find them reasonable and relevant to your core conclusions. All reviewers have issues with the strength and relevance of the claims related to the INCENP domain capable of binding DNA directly and that the binding to α -satellite DNA is slightly more efficient - toning them down as suggested by the refs would seem appropriate in the absence of stronger data. Addressing the reviewers' concerns about transient expression seems needed, and similarly, better and more scholarly discussion of the results in the context of recent and past work is important. Please address all other concerns as well, including if possible Reviewer #2 point #1, although this point seems like a lower priority than the other reviewer comments to us. Please do not hesitate to contact us if you have any questions or wish to discuss the revisions further.

GENERAL GUIDELINES:

Text limits: Character count for an Article is < 40,000, not including spaces. Count includes title page, abstract, introduction, results, discussion, acknowledgments, and figure legends. Count does not include materials and methods, references, tables, or supplemental legends.

Figures: Articles may have up to 10 main text figures. Figures must be prepared according to the policies outlined in our Instructions to Authors, under Data Presentation, <http://jcb.rupress.org/site/misc/ifora.xhtml>. All figures in accepted manuscripts will be screened prior to publication.

IMPORTANT: It is JCB policy that if requested, original data images must be made available. Failure to provide original images upon request will result in unavoidable delays in publication. Please ensure that you have access to all original microscopy and blot data images before submitting your revision.

Supplemental information: There are strict limits on the allowable amount of supplemental data. Articles may have up to 5 supplemental figures. Up to 10 supplemental videos or flash animations are allowed. A summary of all supplemental material should appear at the end of the Materials and methods section.

The typical timeframe for revisions is three months; if submitted within this timeframe, novelty will not be reassessed at the final decision. Please note that papers are generally considered through only one revision cycle, so any revised manuscript will likely be either accepted or rejected.

Thank you for this interesting contribution to the Journal of Cell Biology. You can contact us at the journal office with any questions, cellbio@rockefeller.edu or call (212) 327-8588.

Sincerely,

Yixian Zheng, PhD
Monitoring Editor, Journal of Cell Biology

Melina Casadio, PhD
Senior Scientific Editor, Journal of Cell Biology

Reviewer #1 (Comments to the Authors (Required)):

The chromosomal passenger complex (CPC), composed of Aurora B, INCENP, borealin, and survivin, localizes to centromeres during early mitosis to regulate kinetochore-microtubule attachment, and to the spindle midzone during late mitosis to promote cytokinesis. Multiple mechanisms behind centromere enrichment of the CPC have been reported; Survivin-H3pT3 interaction, Borealin-nucleosome interaction, Borealin-Sgo interaction, and coalescence of the Borealin disordered segment. For the spindle midzone localization, MKLP2-CPC interaction, which is suppressed by

Cdk-dependent phosphorylation of INCENP at T59 and MKLP2 at S532, T857, S867 and S878, plays an important role. However, the mechanism by which MKLP2 interacts with the CPC remained unclear. In this manuscript, Serena et al. report that a novel RRKKRR-motif in the INCENP N-terminal segment plays an important role in the CPC localization during mitosis. By imaging GFP fused INCENP N-terminal fragment in HeLa cells, the authors show that the INCENP RRKKRR-motif is essential for the INCENP N-terminal fragment to localize to centromeres in early mitosis and to spindle midzone in anaphase (Fig. 1, 3). The authors also report that the RRKKRR motif is important for centromere localization of full length INCENP and Aurora B, and for histone H3pS10. X-ray crystallography and ITC analyses using H3pT3 peptide and the complex composed of N-terminal INCENP, survivin, and N-terminal borealin suggest that the INCENP RRKKRR-motif does not affect the survivin-H3pT3 interaction. However, the authors found that the INCENP RRKKRR-motif can bind DNA (Fig. 3). Furthermore, the authors show that interaction of MKLP2 and INCENP in vivo and in vitro depends on the INCENP RRKKRR-motif and MKLP2 amino acid 636-652 region, and that these motifs are critical for spindle midzone localization of INCENP/Aurora B and MKLP2 (Fig 5, 6). Finally, the authors show that INCENP can stimulate the ATPase activity of MKLP2, and this is dampened by phosphorylation of INCENP and MKLP2. Since MKLP2 can compete DNA-binding of INCENP RRKKRR motif, the authors propose that mitotic localization change of the CPC can be modulated by Cdk-dependent control of differential binding of RRKKRR to DNA and MKLP2.

Overall, the discovery that the INCENP RRKKRR motif plays an important role in centromere and spindle midzone localization through facilitating MKLP2 interaction is novel and important. In addition, direct activation of MKLP2 ATPase by the CPC is an exciting result. However, significance of the reported DNA-binding capacity of the RRKKRR motif is not compelling, and it is misleading to state in abstract that the motif binds alpha-satellite DNA. While biochemistry data are generally strong, cytological data are weaker as they are all relied on transient transfection without any quantitative analysis. Below I suggest several specific points to be addressed before publication in JCB.

Major Points

1. All localization analysis of the GFP-INCENP constructs in HeLa cells (Fig 1A, 1B, 3A, 3B, S2E, S2F, S4) are based on transient transfection, and no quantitative analysis has been made. Since GFP tag is often cleaved off upon cellular expression, it is critical to confirm expression of expected products by western blots. Among those localization analyses, Fig 3B, where endogenous INCENP was replaced with RRKKRR mutant, is the key. Quantitative analysis of this experiment must be presented. Details of RNAi and transfection experiment must be disclosed.

In addition, some of conclusions are not consistent with previous observations. For example, it has been reported that Survivin E65A and H80A single point mutants are defective in centromere enrichment but are localized to chromatin (Niedzialkowska et al. MBoC 2012), unlike the apparent defect in chromatin localization reported here. While the authors claim that H80A mutant can be still localized to the centromere, centromeric signals look qualitatively weaker than those in control (Fig S2E). Comparing to Niedzialkowska et al who analyzed cells that stably expressing survivin mutants upon depleting endogenous survivin, the current report is based on transient transfection on top of endogenous proteins. Frankly, I could not find any value of adding Fig. S2E and F over previously published results by Niedzialkowska et al.

2. Although the authors propose that the RRKKRR-motif supports centromere localization of INCENP through binding to alpha-satellite DNA, evidence is far from compelling. Fig 3E shows that the binding affinity is very weak (~20 μ M), and there is no sequence specificity. Alpha-satellite "consensus" DNA used in this study is only a part of the repeat unit, so no positive or negative

conclusion can be made.

The basic amino acids could also show comparable (or perhaps stronger) affinity to other acidic targets, such as RNA, the acidic patch of the nucleosome, and importin alpha. Given a recent report that borealin can directly bind to the nucleosome, acidic patch of H2A and H2B may be the most probable target. At this point, the proposed importance of RRKKRR-alpha-satellite DNA interaction is misleading, and thus summary and interpretation should be adjusted accordingly.

3. One of the major conclusions of this manuscript is that the RRKKRR motif of INCENP is critical for MKLP2 interaction. However, this dependency in cells was only shown by co-IP using overexpressed N-terminal fragment of INCENP (Figure 4). It is critical to demonstrate this dependency in a cell where endogenous INCENP is replaced with a version lacking the RRKKRR motif.

4. Figure 6. It is concerning that CPC80^{ΔRRKKRR} and CPC58 mutants eluted faster than the wild-type version. It would be important to show molecular weights of these mutants analyzed by MALS. Since these mutants also show significant binding to MKLP2, it is possible that the RRKKRR motif does not serve as an interaction site for MKLP2, but helps complex stabilization by an alternative mechanism, for example, by exposing the real MKLP2 binding site. This may explain why CPC mutants eluted faster. Positions of the molecular weight markers should be indicated in the gel images.

5. According to the X-ray crystal structure, MKLP2 636-652 is located on the dimerization domain of the protein (Fig 5S). Therefore, MKLP2 636-652 can be essential for dimerization and/or proper folding of MKLP2. The defects of MKLP2 localization and INCENP interaction by Δ 636-652 may be due to a dimerization/folding defect. The authors should confirm that the deletion of the MKLP2 636-652 does not affect the MKLP2 dimerization and folding, which can be tested with biochemical analysis of recombinant protein such as SEC-MALS.

Minor Points

1. Ainsztein et al have previously reported that INCENP1-68, which contains the RRKKRR motif except for the last R, can be localized to the centromere and the spindle midzone (Ainsztein et al. JCB 1998). This important contribution must be acknowledged.

2. Page 7. "The N-terminus of INCENP of a symmetry related CPC^{ΔNT} molecule projects into the peptide binding site of survivin".

The identical INCENP T3 - Survivin interaction can also be observed in the previously reported structure (Jeyaprasakash et al., Cell, 2007). The authors should mention this.

3. Page 9, line 1-2. The authors describe, "For CPC 58, we identified an additional hydrogen bond between peptide Gln5 and survivin Glu65 side chain (Figure 2A and 2B)."

However, in this structure, the Gln5 main chain does not fit well in the electron density (Fig S1B). Since it is common that fitting software makes an error at the ends of peptide, it is worth checking the structure to conform if the place and angle of the Gln5 main chain are correct, and the statement is accurate.

4. Fig. 7E. Please define how relative ATPase activities were determined.

Reviewer #2 (Comments to the Authors (Required)):

The chromosome passenger complex (CPC), a 4-subunit complex incorporating the Aurora B kinase, INCENP, Borealin, and Survivin subunits, plays crucial functions in chromosome segregation and is active both in mitosis and after anaphase. The localization of the CPC varies during the cell cycle, with an initial recruitment to chromosomes as cells enter mitosis, subsequent enrichment at the centromere/inner kinetochore, and finally transfer to the central spindle upon mitotic exit. How this localization is achieved and regulated is a question of considerable importance, because localized Aurora B activity is probably key to the control of the processes that this kinase regulates. In this study, Michela Serena and co-worker revisit this important question, adding an important new piece to the localization puzzle.

The authors re-examined the requirements for CPC localization to the chromosomes, centromeres, and central spindle/spindle midzone. These experiments led the authors to correct a previous report (Klein et al. 2006) identified the first 58 residues of INCENP as being sufficient to reconstitute a localization module of the CPC together with Borealin and Survivin. The authors show that INCENP(1-58) fails to promote robust localization to any of the three main locations discussed above. Conversely, residues 1-80 of INCENP are sufficient for localization to all three regions. The INCENP extension contains a poly-basic motif, whose function in CPC localization had not been previously analyzed, and that the authors demonstrate to be important both for chromosome and centromere localization before anaphase, and for an interaction with the MKLP2 kinesin required for central spindle localization after anaphase. The authors provide evidence that this region of INCENP can interact with DNA and with MKLP2, and propose that a competition mechanism is at base of the relocation of the CPC from chromosomes and centromeres to the central spindle.

Importantly, the study provides several additional original observations, including crystal structures of the entire CPC localization module with the phosphorylated histone H3 tail, an interaction that promotes chromosome recruitment of the CPC, and the important observation that the CPC may activate the ATPase activity of MKLP2, an observation that was not characterized further but that opens important new directions. Furthermore, the study has a quantitative flare that makes its conclusions quite convincing.

Collectively, I am enthusiastic about the study and I find the conclusions, for the most part, compelling. I strongly recommend publication of this study, but I would like to ask the authors to consider the following points:

1. The phosphorylation of Thr59 of INCENP by CDK1 is crucial for the regulation of INCENP localization, and previous evidence and additional evidence in this study indicate that this phosphorylation ultimately controls the interaction switch that allows INCENP to move to the central spindle through MKLP2. The authors do not analyze the role of this phosphorylation in great detail, but I would like them to consider the following point. When phosphorylated, T59 will position a negative charge in the proximity of the poly-basic motif proposed to bind DNA. There is no attempt to exclude the possibility that the phosphorylation may affect the charge distribution required to bind DNA, and I feel that the authors should at least try to demonstrate that mutation of T59 to E does not affect DNA binding in their assay (if they could phosphorylate the sample with CDK1, it would of course make for a cleaner experiment).

2. There is an insistence on the fact that INCENP promotes binding to alpha-satellite DNA (also in the abstract) but the data sustaining this claim are scanty (very small differences in binding affinity relative to non-alpha-satellite sequences). I understand that this idea reflects the willingness of the authors to explain what features of the CPC promote centromere localization, but in the absence of

stronger evidence, I would tune down this point and just report that there is binding to DNA. The authors are certainly aware that alpha-satellite DNA is not particularly relevant for centromere specification.

3. In the Introduction, I am slightly puzzled by the way the authors report previous work on the role of Borealin in centromere and chromosome recruitment. Abad et al. JCB 2019 show that various perturbations within a "loop" region in Borealin prevents localization not only to chromosome arms as stated, but also to centromeres. As the authors here do not contradict these findings (nor perform binding assays with nucleosomes, unlike Abad et al. 2019), I feel that they should report them for what they show. I add that the new data shown here do not imply that the Abad et al. 2019 paper is incorrect. The two binding mechanisms may very well co-exist, each contributing substantial binding affinity required for centromere localization.

4. On the other hand, the authors do not seem to discuss the implications of their observations for a recent phase separation (PS) model for the localization of the CPC (Trivedi et al. NCB 2019) that their data shatter into pieces. First, the authors of the PS paper assumed that INCENP(1-59) mediates robust centromere localization, an assumption that the authors prove wrong. Second, Trivedi et al. NCB 2019 proposed that initial binding of CPC at centromeres "seeds" phase separation and accretion there (see the model in Figure 7G of the Trivedi et al 2019 paper). Here, Serena and co-workers present observations that are inconsistent with this model. Under conditions in which the endogenous CPC is expressed (and therefore presumably under conditions in which a seed for phase separation at the centromere can be formed), Serena et al. demonstrate that CPC localization modules previously shown in the Trivedi et al. paper to phase separate are unable to reach the centromere, arguing rather strongly that an already existing pool of centromeric CPC is insufficient to drive the recruitment of a phase separating CPC localization module, and rather that features required to bind to specific target sites there are required. As these observations contradicts a major tenet of the Trivedi et al. NCB 2019 paper, I feel that it is important that the field is informed of this inconsistency and I would like to recommend that the authors include it in their discussion.

5. Finally, the authors could elect to cite recent work by Franz Herzog and colleagues (Fischböck-Halwachs et al. eLife 2019) identifying interactions of the CPC within the kinetochore (in *S. cerevisiae*).

Minor points:

Abstract: "This interaction promotes..." In the previous sentence, the authors report two interactions. "This" is ambiguous.

Page 8: "This confers limits..." Please rephrase

Page 16: "...explaining why this is not transported...". Again, "this" is ambiguous: does it refer to MKLP2 or to INCENP(1-58/T59E)?

Figure 1A: please define the "f" sign in the labels for the last two rows.

Figure 1C: There is an almost perfect consensus site for Aurora B at the end of the poly-basic motif. Worth mentioning it?

Figure 5B: In the main text, while discussing this figure, the authors gloss over the problem that their

biochemical and localization analyses don't entirely fit, as the residual interactions observed with the $\Delta 690-705$ and RRSQR mutants are insufficient for their localization, which is as impaired as it is for the $\Delta 636-652$ mutant. The authors should point this out in the text.

Figure 6: I am unclear: do the elution volumes of the same biochemical species change for every run? As presented, this seems to be the puzzling conclusion.

Reviewer #3 (Comments to the Authors (Required)):

Referee report

Molecular basis for MKLP2-dependent Aurora B transport from chromatin to the anaphase central spindle

by Michela Serena, Ricardo Nunes Bastos*, Paul R. Elliott, Francis A. Barr

The paper set to understand the molecular mechanism underlying the CPC localisation to centromere in early mitosis and then to the central spindle after anaphase onset.

The Authors start with the analyses of different INCENP mutants in a series of overexpression experiments to narrow down potential novel regions of the protein involved in its centromere and spindle targeting.

This is followed by a nice study on the crystal structure of the partial CPC complex where, for the first time, a structure of three complex members bound to the H3T3phospho was obtained. From that point the authors move to the analyses of potential additional chromatin binding domains and discover that INCENP is capable of binding chromatin (with a slight preference for satellite DNA). This partly explains the targeting to the centromere.

For the relocalisation to the spindle, the authors address the interplay between INCENP, its chromatin binding domain and MKLP2. They provide evidence that MKLP2 competes for the chromatin binding domain of INCENP and that phosphorylations regulate these mutually exclusive bindings, thus providing an explanation for the re-localisation of the CPC from the centromere in metaphase to the central spindle at anaphase onset.

Overall the study is novel, it expands the current knowledge on the field, it is well conducted and the conclusions mostly supported by the data provided.

However, a few aspects and clarification should be added, including referring to previous known aspect of the CPC biology.

Specific Comments:

Major

1_ Previous studies have already looked at different domains of INCENP and its targeting to the centromere/spindle. In one study in particular, the same domain identified by the authors had already been shown to be important. This work need to be referred to : J Cell Biol. 1998 Dec 28; 143(7): 1763-1774. doi: 10.1083/jcb.143.7.1763 "Randomization of the order of amino acid residues 52-62 in INCENP1-405 (52-62r):GFP specifically abolished this transfer"

2_ Most of the studies have been conducted as overexpression experiments (apart from Figure 3B). In order to avoid incorrect conclusions due to the complex interactions between the endogenous proteins and the overexpressed one, the author should provide the localization of the mutants (at least the major constructs T59A and T59B) in an RNAi background.

3_ The Authors show that INCENP contains a domain capable of binding DNA directly and that the binding to α -satellite DNA is slightly more efficient.

These differences in vitro are really small and, although several weak interactions could sum up in vivo and provide an overall strong interaction, there are some caveats that would need to be examined.

One important aspect to consider is the fact that the CPC can well accumulate and function at the centromere of chicken chromosomes that do not contain α -satellite and are not composed of repetitive sequences (<http://dx.doi.org/10.1016/j.devcel.2013.02.009>). If the hypothesis provided was correct, then these centromeres should accumulate less CPC or less stable one. However, the error correction mechanisms and the segregation defects are the same as for centromere with α -satellite or repetitive DNA. Therefore the small difference observed in vitro may not be of relevance in cells.

These factors should be discussed and taken into account.

4_ The Authors hypothesis is that the INCENP RRKKRR region is responsible for both binding the DNA and MKLP2 and also that binding to INCENP is necessary for MKLP2 localisation to the spindle. In this context the author should explain why in INCENP 1-52 (which does not contain that motif) MKLP2 does normally localizes to the spindle (Figure 1 A). Maybe in an RNAi background would not?

5_ How the Author explain the sequence of events for the transfer of the CPC at anaphase onset? The H3 de-phosphorylation occurs later than the transfer of the CPC to the spindle. An idea of the sequential de-phosphorylation and the strengths of interactions would be important to understand the correct mechanism.

Minor:

1_ Page 4:

"A counteracting phosphatase PP1-repoman inhibited by CDK1-cyclin B then dephosphorylates H3pT3 during mitotic exit (Qian et al., 2015; Trinkle-Mulcahy et al., 2006; Vagnarelli et al., 2006)."

The references are not correct. Qian 2015 does not demonstrate the H3T3 dephosphorylation. The correct quotes are Qian et al, 2011 and Vagnarelli et al, 2011.

2_ Page 8

"This confers limits the position of the N-terminus of the peptide with respect to the phospho-binding pocket, and explains the selectivity for..."

This sentence is not clear

3_ Page 15

"There are two non-exclusive possibilities: MKLP2 binding either competes with phospho-histone binding or DNA binding. Comparison of the binding affinity for phosphorylated and non-phosphorylated histone H3 by the MKLP2:CPC80 complex was then performed by isothermal titration calorimetry."

This sentence is not clear

Reviewer #1 (Comments to the Authors (Required)):

Overall, the discovery that the INCENP RRKKRR motif plays an important role in centromere and spindle midzone localization through facilitating MKLP2 interaction is novel and important. In addition, direct activation of MKLP2 ATPase by the CPC is an exciting result. However, significance of the reported DNA-binding capacity of the RRKKRR motif is not compelling, and it is misleading to state in abstract that the motif binds alpha-satellite DNA. While biochemistry data are generally strong, cytological data are weaker as they are all relied on transient transfection without any quantitative analysis. Below I suggest several specific points to be addressed before publication in JCB.

We thank the referee for their comments which we have attempted to address to the best of our ability with new data and revision to the text as suggested in their review.

Major Points

1. All localization analysis of the GFP-INCENP constructs in HeLa cells (Fig 1A, 1B, 3A, 3B, S2E, S2F, S4) are based on transient transfection, and no quantitative analysis has been made. Since GFP tag is often cleaved off upon cellular expression, it is critical to confirm expression of expected products by western blots. Among those localization analyses, Fig 3B, where endogenous INCENP was replaced with RRKKRR mutant, is the key. Quantitative analysis of this experiment must be presented. Details of RNAi and transfection experiment must be disclosed.

To address these concerns, we have produced stable doxycycline inducible cell lines with single integrated copies of the INCENP wild type and mutant transgenes. Western blots are used to confirm expression level and show the GFP-tag was not cleaved from the protein as the reviewer suggested might be the case. We then use these cell lines in the revised figures for both INCENP localisation and functional experiments. Quantifications of these data are presented in the figures (see Figure 1E, 1F and 9B). All experimental details are present in the methods section and figure legends contain details of the quantitative statistics.

In addition, some of conclusions are not consistent with previous observations. For example, it has been reported that Survivin E65A and H80A single point mutants are defective in centromere enrichment but are localized to chromatin (Niedzialkowska et al. MBoC 2012), unlike the apparent defect in chromatin localization reported here. While the authors claim that H80A mutant can be still localized to the centromere, centromeric signals look qualitatively weaker than those in control (Fig S2E). Comparing to Niedzialkowska et al who analyzed cells that stably expressing survivin mutants upon depleting endogenous survivin, the current report is based on transient transfection on top of endogenous proteins. Frankly, I could not find any value of adding Fig. S2E and F over previously published results by Niedzialkowska et al.

We have revised and extended this data to include an analysis of the role of metaphase CPC targeting in the MKLP2-dependent transport process in anaphase cells (new Figure 7). To do this, we have carefully repeated these experiments using stable cell lines depleted for endogenous survivin, which as the referee points out will give more reproducible control of expression levels. Western blot confirms the levels of GFP-survivin, and mutants thereof, are equivalent to the endogenous survivin protein (Figure S5A). We can confirm the previous result that H80A spreads out away from centromeres on to chromatin (Figure 7A) (Niedzialkowska et al. MBoC 2012). In addition, we added two other mutants (K62A and K62A-H80A) which according to our structure, perturb H3pT3 recognition. These, like H80A, show a more diffuse localisation on chromatin in metaphase, rather than a precise centromeric distribution observed with WT-Survivin. In anaphase all three mutants localises to the central spindle (Figure 7B).

We then extend this by asking if histone H3 backbone binding is crucial for anaphase targeting and MKLP2 interaction. To do this we use the E65A mutant, alone and in combination with H80A. E65A is important for recognition of the R2 in histone H3, and is cytoplasmic in both metaphase and anaphase (Figure 7A and 7B).

Even if some of these data are partially presented in previous publications, here we wanted to address the specific point regarding anaphase localisation and MKLP2 binding, which has not been adequately investigated before. Our data show that CPC mutants, defective for chromatin

binding (survivin E65A or E65A-H80A), are unable to localise at the central spindle in anaphase, even if they are still able to form a complex with MKLP2 *in vitro* (Figure 7C).

This leads us to an important conclusion. MKLP2 binding to the CPC is restricted to the pool of CPC present on chromatin in metaphase. The behaviour of the K62A and H80A mutants supports the view that centromere localisation is less crucial than general chromatin binding. Furthermore, it supports our hypothesis that an intermediate state exists, where MKLP2 is able to bind CPC when it's still on centromeres, competing with DNA for RRKKRR motif binding on INCENP without disrupting histone binding.

Thus, if the CPC is not properly localising in the right place at the metaphase to anaphase transition, it cannot be picked-up by MKLP2.

2. Although the authors propose that the RRKKRR-motif supports centromere localization of INCENP through binding to alpha-satellite DNA, evidence is far from compelling. Fig 3E shows that the binding affinity is very weak (~20 μM), and there is no sequence specificity. Alpha-satellite "consensus" DNA used in this study is only a part of the repeat unit, so no positive or negative conclusion can be made.

We have removed specific claims about α -satellite DNA from the abstract and modified the text in response to comments by another reviewer. The major question we ask in these experiments is if the RRKKRR-motif can interact with DNA. To address this, we have used two methods to investigate DNA binding: electrophoretic mobility shift assays with linear plasmid DNA (EMSA) and the more quantitative microscale thermophoresis with defined DNA duplexes to assign K_d values. These highly reproducible data are described in the new Figure 3, then extended in for complexes with the bound MKLP2 fragment in Figure 8B and 8C. For MST the original data is available as a supplemental spreadsheet in Table S2. Both methods indicate DNA binding in the μM range for the wild type complexes. The effects of mutation in the RRKKRR motif are not small, with both EMSA and MST reporting a large change in binding affinity.

We now clearly state there is an ~1.25-fold difference in the K_d in favour of α -satellite DNA. This is also raised in the revised discussion as a point needing further investigation in the context of nucleosome binding, since in cells DNA binding will be restricted by nucleosome position.

We therefore favour the idea that multiple weak interactions in the μM range promote CPC localisation, and as explained in our introduction and discussion the literature is consistent with this proposal. Because the CPC shows dynamic localisation and has to be removed from chromatin, a single site high affinity mode of interaction is unlikely. Supporting this statement, the reported histone H3 / H3pT3 binding is in the range of 2-10 μM (our work and other studies, and we see DNA interactions with K_d of ~24 μM by MST with short DNA duplexes or estimated to be ~3 μM in EMSA with a longer template DNA.

The basic amino acids could also show comparable (or perhaps stronger) affinity to other acidic targets, such as RNA, the acidic patch of the nucleosome, and importin alpha. Given a recent report that borealin can directly bind to the nucleosome, acidic patch of H2A and H2B may be the most probable target. At this point, the proposed importance of RRKKRR-alpha-satellite DNA interaction is misleading, and thus summary and interpretation should be adjusted accordingly. The reviewer proposes that the RRKKRR-motif may bind to the basic region of histone H2A/H2B (at the face of the nucleosome), some other protein or RNA, and not only DNA. We don't have any data on this, so cannot exclude these possibilities. It would require reconstitution with intact nucleosomes and further structural studies to address the specific point made about the acidic patch on H2A and H2B. We aim to do this, but it goes beyond our current study. We do as explained, present data on DNA-binding and characterise specific mutants. This and the idea the reviewer mentions are more clearly described in the revised discussion.

3. One of the major conclusions of this manuscript is that the RRKKRR motif of INCENP is critical for MKLP2 interaction. However, this dependency in cells was only shown by co-IP using overexpressed N-terminal fragment of INCENP (Figure 4). It is critical to demonstrate this

dependency in a cell where endogenous INCENP is replaced with a version lacking the RRKKRR motif.

The revised Figure 9A and 9B show the experiment the reviewer requests. This was performed using stable cell lines expressing inducible copies of full-length INCENP or the full-length RRKKRR-mutant.

4. Figure 6. It is concerning that CPC80^{RRKKRR} and CPC58 mutants eluted faster than the wild-type version. It would be important to show molecular weights of these mutants analyzed by MALS. Since these mutants also show significant binding to MKLP2, it is possible that the RRKKRR motif does not serve as an interaction site for MKLP2, but helps complex stabilization by an alternative mechanism, for example, by exposing the real MKLP2 binding site. This may explain why CPC mutants eluted faster. Positions of the molecular weight markers should be indicated in the gel images.

The SEC-MALS data was not corrected for the 2ml fraction collector delay with respect to the UV measurement. This has been corrected in the revised figure 6, and molecular weight markers have been added to the inset gel panels.

CPI58 and the CPI80^{RRKKRR} do not form SEC stable complexes with MKLP2 (Figure 6B and 6C). Additionally, they do not co-IP with MKLP2 (Figure 4) or localise correctly in cells (Figure 1 and S1). If the idea proposed by the referee was correct, then CPI58 which lacks the RRKKRR-motif should expose an MKLP2 binding site and thus interact with MKLP2. However, this is not the case. The structures we present in Figure 2 and S2 also speak against this idea, since they have effectively identical surfaces.

5. According to the X-ray crystal structure, MKLP2 636-652 is located on the dimerization domain of the protein (Fig 5S). Therefore, MKLP2 636-652 can be essential for dimerization and/or proper folding of MKLP2. The defects of MKLP2 localization and INCENP interaction by Δ 636-652 may be due to a dimerization/folding defect. The authors should confirm that the deletion of the MKLP2 636-652 does not affect the MKLP2 dimerization and folding, which can be tested with biochemical analysis of recombinant protein such as SEC-MALS.

To address this question, we performed SEC with GFP-MKLP WT and GFP-MKLP2 Δ 636-652. The collected fractions were checked by western blot, both with anti-MKLP2 and anti-GFP antibodies. The results indicate that there is no difference in the elution volume between MKLP2 WT and Δ 636-652, which are also co-eluted together with endogenous MKLP2. In addition, the estimated molecular weight (calculated by column calibration with standard proteins) corresponds to approximately ~220 KDa consistent with a dimer architecture.

Minor Points

1. Ainsztein et al have previously reported that INCENP1-68, which contains the RRKKRR motif except for the last R, can be localized to the centromere and the spindle midzone (Ainsztein et al. JCB 1998). This important contribution must be acknowledged.

This citation has been added to the text. We have also revised the relevant sections of the results to refer to this work.

2. Page 7. "The N-terminus of INCENP of a symmetry related CPC^{ANT} molecule projects into the peptide binding site of survivin".

The identical INCENP T3 - Survivin interaction can also be observed in the previously reported structure (Jeyaprasath et al., Cell, 2007). The authors should mention this.

To our knowledge, the authors of the previously reported structure (Jeyaprasath et al., Cell, 2007) did not comment on this interaction in their manuscript and we were unable to find any mention of this point in other literature. We agree with the referee that the same crystal contact between the symmetry-related N-terminus INCENP and the peptide binding site of survivin is observed in that structure, and have adjusted the results text accordingly.

3. Page 9, line 1-2. The authors describe, "For CPC 58, we identified an additional hydrogen bond between peptide Gln5 and survivin Glu65 side chain (Figure 2A and 2B)." However, in this structure, the Gln5 main chain does not fit well in the electron density (Fig S1B).

Since it is common that fitting software makes an error at the ends of peptide, it is worth checking the structure to conform if the place and angle of the Gln5 main chain are correct, and the statement is accurate.

We have checked the structure fits in the density map in Coot as requested. The hydrogen bond is to the peptide backbone and we have revised this in the text.

4. Fig. 7E. Please define how relative ATPase activities were determined.

This is now explained in the figure legend. Briefly, the comparison is the rate at the 15min time point from the kinetic analysis. This is a simpler way to compare different mutants to the wild type CPC.

Reviewer #2 (Comments to the Authors (Required)):

Collectively, I am enthusiastic about the study and I find the conclusions, for the most part, compelling. I strongly recommend publication of this study, but I would like to ask the authors to consider the following points:

We have carefully considered the points provided by the referee and added new data to address their key concerns. We have also made careful revisions to the text moderate the statements on DNA binding and sequence specificity, and added more discussion of this point.

1. The phosphorylation of Thr59 of INCENP by CDK1 is crucial for the regulation of INCENP localization, and previous evidence and additional evidence in this study indicate that this phosphorylation ultimately controls the interaction switch that allows INCENP to move to the central spindle through MKLP2. The authors do not analyze the role of this phosphorylation in great detail, but I would like them to consider the following point. When phosphorylated, T59 will position a negative charge in the proximity of the poly-basic motif proposed to bind DNA. There is no attempt to exclude the possibility that the phosphorylation may affect the charge distribution required to bind DNA, and I feel that the authors should at least try to demonstrate that mutation of T59 to E does not affect DNA binding in their assay (if they could phosphorylate the sample with CDK1, it would of course make for a cleaner experiment).

To address this point, we repeated both EMSA and MST assays with a CPI80 molecule with a phospho-mimetic T59E mutation. The results presented in Figure 3G indicate that CPC80^{T59E} is able to bind DNA like the wild type protein in EMSA. The affinity derived by MST is slightly reduced compared to the wild type CPI80, but the K_d was within a factor of two. Other binding defective constructs such as CPI80^{RRKKRR} showed no binding and we were unable to calculate K_d values.

To carry out the experiment with CDK-phosphorylation would require stoichiometric modification, which is difficult to achieve in vitro.

2. There is an insistence on the fact that INCENP promotes binding to alpha-satellite DNA (also in the abstract) but the data sustaining this claim are scanty (very small differences in binding affinity relative to non-alpha-satellite sequences). I understand that this idea reflects the willingness of the authors to explain what features of the CPC promote centromere localization, but in the absence of stronger evidence, I would tune down this point and just report that there is binding to DNA. The authors are certainly aware that alpha-satellite DNA is not particularly relevant for centromere specification.

We have removed claims about α -satellite DNA from the abstract and simply refer to DNA binding.

We should also note that the K_d value calculated by MST in 100 mM NaCl containing solution does not reflect what the real K_d may be in the cells, where the CPC is bound to nucleosomes and the interplay with many other charge molecules will determine the "true" binding affinity. Our minimal model is simplified by using a 40-nt DNA fragment, and the number presented here has to be considered as a measure of binding under defined conditions. All these factors are likely to be relevant for CPC recognition of chromosome arms and centromeres in cells.

The data presented in Figure 3 is from multiple EMSA assays and MST experiments. The graph reported in Figure 3D, show mean \pm SEM of the fluorescence values (n=4). The average value K_d was calculated using the NanoTemper analysis software. A statistical analysis using two-way ANOVA test was performed to measure the significance of the difference in values observed for centromeric α -satellite and non α -satellite DNA. Anyway, for the reasons explained in the above point 1, we moderated our interpretation accordingly.

3. In the Introduction, I am slightly puzzled by the way the authors report previous work on the role of Borealin in centromere and chromosome recruitment. Abad et al. JCB 2019 show that various perturbations within a "loop" region in Borealin prevents localization not only to chromosome arms as stated, but also to centromeres. As the authors here do not contradict these findings (nor perform binding assays with nucleosomes, unlike Abad et al. 2019), I feel that

they should report them for what they show. I add that the new data shown here do not imply that the Abad et al. 2019 paper is incorrect. The two binding mechanisms may very well co-exist, each contributing substantial binding affinity required for centromere localization.

We certainly don't dispute any finding from the Abad study, in fact we feel that it sets the direction for our own future work where we would like to reconstitute the CPC on to nucleosomes. We have revised the introduction text as follows:

"Borealin dimerises through a structured domain at the C-terminus (Bekier et al., 2015; Bourhis et al., 2009) and makes directly and specific contact to nucleosomes (Abad et al., 2019). These properties will increase the avidity of the CPC for chromatin. Centromere specific enrichment in mitosis is promoted by CDK1-phosphorylation of an unstructured region of borealin upstream of the dimerization domain, which promotes interaction with the centromeric protein shugoshin (Tsukahara et al., 2010)."

4. On the other hand, the authors do not seem to discuss the implications of their observations for a recent phase separation (PS) model for the localization of the CPC (Trivedi et al. NCB 2019) that their data shatter into pieces. First, the authors of the PS paper assumed that INCENP(1-59) mediates robust centromere localization, an assumption that the authors prove wrong. Second, Trivedi et al. NCB 2019 proposed that initial binding of CPC at centromeres "seeds" phase separation and accretion there (see the model in Figure 7G of the Trivedi et al 2019 paper). Here, Serena and co-workers present observations that are inconsistent with this model. Under conditions in which the endogenous CPC is expressed (and therefore presumably under conditions in which a seed for phase separation at the centromere can be formed), Serena et al. demonstrate that CPC localization modules previously shown in the Trivedi et al. paper to phase separate are unable to reach the centromere, arguing rather strongly that an already existing pool of centromeric CPC is insufficient to drive the recruitment of a phase separating CPC localization module, and rather that features required to bind to specific target sites there are required. As these observations contradict a major tenet of the Trivedi et al. NCB 2019 paper, I feel that it is important that the field is informed of this inconsistency and I would like to recommend that the authors include it in their discussion.

We have not looked at phase separation and therefore cannot comment directly on the phase separation behaviour described in the Trivedi paper. We agree with the reviewer that our data and other published work do not provide strong support for the phase separation driven targeting model.

5. Finally, the authors could elect to cite recent work by Franz Herzog and colleagues (Fischböck-Halwachs et al. eLife 2019) identifying interactions of the CPC within the kinetochore (in *S. cerevisiae*).

This is an interesting study and one that we would definitely discuss if writing a review on the topic of centromere architecture and CPC-targeting in different organisms. The relevance for our work at present isn't clear and would require more studies looking at larger assemblies with nucleosomes and potentially centromeric proteins.

Minor points:

Abstract: "This interaction promotes..." In the previous sentence, the authors report two interactions. "This" is ambiguous.

Page 8: "This confers limits..." Please rephrase

The text has been rewritten.

Page 16: "...explaining why this is not transported...". Again, "this" is ambiguous: does it refer to MKLP2 or to INCENP(1-58/T59E)?

Figure 1A: please define the "f" sign in the labels for the last two rows.

This is defined in the figure legends; *f* = full-length construct, not a truncation.

Figure 1C: There is an almost perfect consensus site for Aurora B at the end of the poly-basic

motif. Worth mentioning it?

We have noticed the same feature, but have no evidence it is phosphorylated.

Figure 5B: In the main text, while discussing this figure, the authors gloss over the problem that their biochemical and localization analyses don't entirely fit, as the residual interactions observed with the $\Delta 690-705$ and RRSQR mutants are insufficient for their localization, which is as impaired as it is for the $\Delta 636-652$ mutant. The authors should point this out in the text.

The MKLP2 $\Delta 690-705$ deletion and RRSQR mutant localise like the wild-type protein to the central spindle, however Aurora B transport appears slightly defective in the RRSQR mutant. We have added a more complete explanation to the text. This matches the behaviour in the IP (Figure 5A), where the RRSQR mutant shows reduced CPC co-IP.

Figure 6: I am unclear: do the elution volumes of the same biochemical species change for every run? As presented, this seems to be the puzzling conclusion.

The SEC-MALS data was not corrected for the fraction collector delay of 2ml. This has now been in the revised Figure 6.

Reviewer #3 (Comments to the Authors (Required)):

Referee report

Overall the study is novel, it expands the current knowledge on the field, it is well conducted and the conclusions mostly supported by the data provided. However, a few aspects and clarification should be added, including referring to previous known aspect of the CPC biology.

Major comments:

1_Previous studies have already looked at different domains of INCENP and its targeting to the centromere/spindle. In one study in particular, the same domain identified by the authors had already been shown to be important. This work need to be referred to : J Cell Biol. 1998 Dec 28; 143(7): 1763-1774. doi: 10.1083/jcb.143.7.1763 "Randomization of the order of amino acid residues 52-62 in INCENP1-405 (52-62r):GFP specifically abolished this transfer"

We have revised the text to better explain the background to our study and cite the Ainsztein et al 1998 paper referred to by the reviewer. The region they permutate in the study is the disordered loop containing the CDK-site and lies immediately adjacent to the RRKKRR-motif. We note that the change made by Ainsztein moves the negative charge in the sequence towards the RRKKRR-motif, similar to a T59E mutant, and is therefore likely to attenuate binding to MKLP2 in a similar manner.

"The search for the MKLP2 binding region on the CPC can be narrowed down to the N-terminal region of INCENP for two reasons (Figure 1A). First, phosphorylation at T59 has been shown to prevent CPC transport (Hummer and Mayer, 2009). Second, the first 68 amino acids of INCENP have been reported to support localisation to the anaphase spindle (Ainsztein et al., 1998)."

2_Most of the studies have been conducted as overexpression experiments (apart from Figure 3B). In order to avoid incorrect conclusions due to the complex interactions between the endogenous proteins and the overexpressed one, the author should provide the localization of the mutants (at least the major constructs T59A and T59B) in an RNAi background.

To address these concerns, we have produced stable doxycycline inducible cell lines with single integrated copies of the INCENP wild type and mutant transgenes. Western blots are used to confirm expression level and show the GFP-tag was not cleaved from the protein as the reviewer suggested might be the case. We then use these cell lines in the revised figures for both INCENP localisation and functional experiments. Quantifications of these data are presented in the figures (see Figure 1E, 1F and 9B). All experimental details are present in the methods section and figure legends contain details of the quantitative statistics.

3_The Authors show that INCENP contains a domain capable of binding DNA directly and that the binding to α -satellite DNA is slightly more efficient. These differences in vitro are really small and, although several weak interactions could sum up in vivo and provide an overall strong interaction, there are some caveats that would need to be examined.

Many biological systems create specificity or regulation by combining multiple weak interactions. The classic example is the immune system. In this instance, we have a complex the CPC which can make multiple weak interactions with the repetitive chromatin making up the surface of the chromosome.

One important aspect to consider is the fact that the CPC can well accumulate and function at the centromere of chicken chromosomes that do not contain α -satellite and are not composed of repetitive sequences (<http://dx.doi.org/10.1016/j.devcel.2013.02.009>). If the hypothesis provided was correct, then these centromeres should accumulate less CPC or less stable one. However, the error correction mechanisms and the segregation defects are the same as for centromere with α -satellite or repetitive DNA. Therefore, the small difference observed in vitro may not be of relevance in cells.

Our work is done in human cells and is to a large extent focussed on the mechanism of CPC transport to the central spindle. The reviewer speculates about what our results might mean in chicken, but we don't feel it is appropriate for us to do the same.

4_ The Authors hypothesis is that the INCENP RRKKRR region is responsible for both binding the DNA and MKLP2 and also that binding to INCENP is necessary for MKLP2 localisation to the spindle. In this context the author should explain why in INCENP 1-52 (which does not contain that motif) MKLP2 does normally localizes to the spindle (Figure 1 A). Maybe in an RNAi background would not?

This appears to be an error on the part of the reviewer. We don't use an INCENP 1-52 construct, so we assume this comment refers to INCENP 1-58. Because we use a wild type background, MKLP2 will localise through interaction with the endogenous CPC.

As requested, we have added data on the localisation in the RNAi background in stable inducible cell lines (see Figure S1C-S1E). In this background, MKLP2 fails to localise to the anaphase spindle in the INCENP 1-58, RRKKRR motif and T59 mutants (Figure S1D). By contrast, MKLP2 does localise when wild type INCENP or the 1-80 fragment are used (Figure S1D).

5_ How the Author explain the sequence of events for the transfer of the CPC at anaphase onset? The H3 de-phosphorylation occurs later than the transfer of the CPC to the spindle. An idea of the sequential de-phosphorylation and the strengths of interactions would be important to understand the correct mechanism.

Since MKLP2-CPC is still able to bind H3pT3, but not DNA, we hypothesise a transient intermediate state in the transport model (Figure 10). We provide affinity measurements for Histone H3 and DNA binding, and these inform our proposed mechanism. As described in the introduction, published work reports dephosphorylation for INCENP T59, Histone H3T3 and MKLP2.

Minor comments:

1_ Page 4:

"A counteracting phosphatase PP1-repoman inhibited by CDK1-cyclin B then dephosphorylates H3pT3 during mitotic exit (Qian et al., 2015; Trinkle-Mulcahy et al., 2006; Vagnarelli et al., 2006)."

The references are not correct. Qian 2015 does not demonstrate the H3T3 dephosphorylation. The correct quotes are Qian et al, 2011 and Vagnarelli et al, 2011.

The correct citations have been inserted.

2_ Page 8

"This confers limits the position of the N-terminus of the peptide with respect to the phospho-binding pocket, and explains the selectivity for..."

This sentence is not clear.

This text has been revised:

"The histone H3 peptide is positioned within the binding pocket through a series of interactions that explain the sequence recognition and selectivity for the N-terminus. The side chain of the free N-terminal Ala1 is inserted in a small hydrophobic pocket formed by survivin L64 and W67 (Figure 2A and 2B, enlarged regions). This limits the position of the N-terminus of the peptide with respect to the phospho-binding pocket, and explains the selectivity for phosphorylation at the 3-position of the peptide."

3_ Page 15

"There are two non-exclusive possibilities: MKLP2 binding either competes with phospho-histone binding or DNA binding. Comparison

of the binding affinity for phosphorylated and non-phosphorylated histone H3 by the MKLP2:CPC80 complex was then performed by isothermal titration calorimetry."

This sentence is not clear.

This text has been revised:

"From the data presented so far, there are two non-exclusive possibilities that we can test: MKLP2 competes with either phospho-histone binding or DNA binding. First, isothermal titration calorimetry was used to investigate the binding affinity of the MKLP2:CPI80 complex for phosphorylated and non-phosphorylated histone H3."

March 13, 2020

Re: JCB manuscript #201910059R

Prof. Francis A Barr
University of Oxford
Department of Biochemistry
South Parks Road
Oxford OX1 3QU
United Kingdom

Dear Prof. Barr,

Thank you for submitting your revised manuscript entitled "Molecular basis of MKLP2-dependent Aurora B transport from chromatin to the anaphase central spindle". The manuscript has been seen by the original reviewers whose full comments are appended below. While the reviewers continue to be overall positive about the work in terms of its suitability for JCB, some important issues remain.

Please address the remaining comments by revision of the text where appropriate, or with experimental data already in hand, and attend to the following revisions to meet our formatting requirements:

- Provide the main and supplementary texts as separate, editable .doc or .docx files
- Provide main and supplementary figures as separate, editable files according to the instructions for authors on JCB's website paying particular attention to the guidelines for preparing images and blots at sufficient resolution for screening and production
- Provide tables as excel files
- Add paragraph after the Materials and Methods section briefly summarizing all "Online Supplementary Materials"

Our general policy is that papers are considered through only one revision cycle; however, given that the suggested changes are relatively minor we are open to one additional short round of revision. Please note that I will expect to make a final decision without additional reviewer input upon resubmission.

Please submit the final revision within one month, along with a cover letter that includes a point by point response to the remaining reviewer comments.

Thank you for this interesting contribution to the Journal of Cell Biology. You can contact me or the scientific editor listed below at the journal office with any questions, cellbio@rockefeller.edu or call (212) 327-8588.

Sincerely,

Yixian Zheng, Ph.D.

Monitoring Editor

Marie Anne O'Donnell, Ph.D.
Scientific Editor

Journal of Cell Biology

Reviewer #1 (Comments to the Authors (Required)):

The authors have made great efforts and clarified most of my concerns. I recommend publication of the manuscript in JCB after addressing a few minor points listed below.

1. Figures 7A & B. The key point of this experiment is that Survivin E65A-H80A mutant defective in H3 binding and metaphase chromosome localization is also defective in anaphase spindle midzone localization of MKLP-Survivin. This is a new interesting result, but since the mutant fails to show any positive activities in the presented cellular assay, it is important to show that expected GFP-tagged proteins are indeed expressed in this cell line using immunoblotting, as I had emphasized the importance of such analysis in my previous review.

2. In Discussion, the authors stated, "Survivin E65A mutants deficient for histone H3 recognition are unable to localise to chromatin and fail to target to the anaphase spindle or microtubule structures, despite being proficient for MKLP2 binding (Figure 7). This implies that, in cells, MKLP2 binding to the CPC is restricted to the surface of chromatin."

This is one of several possibilities, and there is no evidence indicating that CPC with Survivin E65A mutant is defective in interacting with MKLP2 in anaphase. I also found it difficult to understand how several possible regulatory mechanisms discussed in the following section can explain this model. Regarding the possibility that binding of the N terminus of INCENP to Survivin has a function, I am skeptical since INCENP N terminal sequences are not conserved among vertebrates. This model seems out of context, as it does not explain why Survivin-H3 interaction is important for CPC-MKLP2 interaction. I think this section requires further editing.

3. Page 12. "Importantly, T59E which mimics the mitotic phosphorylation of INCENP does not abolish DNA binding."

This statement is confusing, since the T59E mutation greatly lowers the affinity to DNA. It is not obvious why maintenance of this very weak DNA binding is "important".

4. Although I understand the possible importance of weak multivalent binding, I maintain my skepticism about the authors' claim that this weak INCENP-DNA binding is physiologically meaningful. The authors may want to discuss a possibility that other negatively charged targets on chromatin may be more relevant.

5. Figures 6B and C. In my previous comments, I asked the authors show molecular weights of each peak determined by MALS, expecting that they had collected the data. Unfortunately, the authors did not respond to this request. Although the authors argue that tested CPI mutants do not form stable complex with MKLP2, the elution profile indicates that they indeed interact with MKLP2 by showing an apparent shift of elution peaks, and also disappearance of MKLP2 oligomers. Currently,

it is not clearly stated if the middle peak represents a complex or not, but if I read the data correctly, it contains a complex of CPI mutants and MKLP2. Disclosing the MALS data will greatly help readers understand how to interpret the data. In my opinion, if proteins maintain interaction in SEC, the protein complex is not unstable. So, I feel that it is misleading to state that the CPI mutants do not form a stable complex with MKLP2. It would be better to describe that it forms a less stable complex than the wild-type does.

6. Page 8, line 33. "espectively" should read "respectively".

Reviewer #3 (Comments to the Authors (Required)):

The revised version of the manuscript has been quite significantly improved in several aspects. The Authors have addressed the comments I have raised aside from one point.

It is quite disappointing that the Author have dismissed a comment that is quite important in this context.

"the centromere of chicken chromosomes that do not contain α -satellite and are not composed of repetitive sequences (<http://dx.doi.org/10.1016/j.devcel.2013.02.009>). If the hypothesis provided was correct, then these centromeres should accumulate less CPC or less stable one. However, the error correction mechanisms and the segregation defects are the same as for centromere with α -satellite or repetitive DNA. Therefore, the small difference observed in vitro may not be of relevance in cells.

Our work is done in human cells and is to a large extent focussed on the mechanism of CPC transport to the central spindle. The reviewer speculates about what our results might mean in chicken, but we don't feel it is appropriate for us to do the same."

The comment here is very important for the interpretation of the data. The region of INCENP involved in chromatin binding is exactly the same across species (as presented in Figure 1), the CPC dynamics and function is the same in all vertebrates and there is not a "human specific" or "chicken specific" function.

If the Authors want to make a claim, they have to take into account all the biological evidence and not only the ones that fit their hypothesis.

Since most of the chicken centromeres do not have alpha satellite, this cannot be the reader for INCENP binding to chromatin.

This was also in line with the concerns expressed by other referees.

Although the authors have down-toned the claim in the revised version and contemplate other possibilities, it should become clearer in the text.

Response to reviewer comments

Reviewer #1 (Comments to the Authors (Required)):

The authors have made great efforts and clarified most of my concerns. I recommend publication of the manuscript in JCB after addressing a few minor points listed below.

1. Figures 7A & B. The key point of this experiment is that Survivin E65A-H80A mutant defective in H3 binding and metaphase chromosome localization is also defective in anaphase spindle midzone localization of MKLP-Survivin. This is a new interesting result, but since the mutant fails to show any positive activities in the presented cellular assay, it is important to show that expected GFP-tagged proteins are indeed expressed in this cell line using immunoblotting, as I had emphasized the importance of such analysis in my previous review.

The western blots for survivin requested by the referee were already provided in the revised submission (Figure S5A). This was also referred to in the following text on page 16:

"To do this we investigated the role of histone H3 binding by survivin, by creating structure guided mutations to differentiate the role of H3pT3 binding and H3 backbone binding in targeting to chromatin. Cells expressing these proteins were then depleted of endogenous survivin, and expression level confirmed by western blotting (Figure S5A)."

This concern is fully addressed by Figure S5A and the manuscript text in the relevant context of Figure 7A and 7B (see remainder of page 16).

2. In Discussion, the authors stated, "Survivin E65A mutants deficient for histone H3 recognition are unable to localise to chromatin and fail to target to the anaphase spindle or microtubule structures, despite being proficient for MKLP2 binding (Figure 7). This implies that, in cells, MKLP2 binding to the CPC is restricted to the surface of chromatin."

This is one of several possibilities, and there is no evidence indicating that CPC with Survivin E65A mutant is defective in interacting with MKLP2 in anaphase. I also found it difficult to understand how several possible regulatory mechanisms discussed in the following section can explain this model. Regarding the possibility that binding of the N terminus of INCENP to Survivin has a function, I am skeptical since INCENP N terminal sequences are not conserved among vertebrates. This model seems out of context, as it does not explain why Survivin-H3 interaction is important for CPC-MKLP2 interaction. I think this section requires further editing.

Our data show that the survivin E65A/H80A double mutant can interact with MKLP2 in vitro, yet fails to target to the anaphase spindle in vivo (Figure 7 and S5). We propose in the discussion that MKLP2 can only interact with the active chromatin bound pool of the CPC in cells. This active chromatin bound CPC is the pool created by the survivin-H3 interaction. This is discussed in the context of the pathways known to regulate the Aurora B/CPC either on chromatin or in the cytoplasm. As shown by others, the CPC is sequestered in an inactive state by nucleoplasmin in the cytoplasm. Additionally, there is ubiquitin-dependent regulation of the CPC on chromatin that may play a role promoting anaphase removal in vivo. This addresses the first part of the point being made by the referee, see text on pages 21-22 (new text is underlined):

"This implies that, in cells, MKLP2 binding to the CPC at the onset of anaphase is restricted to sites on chromatin. At present this dependency for chromatin targeting cannot be fully explained, but it does indicate the presence of further regulation of MKLP2 or the CPC. This is conceivably via the known ubiquitin-dependant regulators of Aurora B function at chromatin (Dobrynin et al., 2011; Krupina et al., 2016; Ramadan et al., 2007). Other work shows that inactive cytoplasmic CPC is chaperoned by the nucleoplasmin family proteins (Hanley et al., 2017), suggesting that this might inhibit binding to other factors such as MKLP2 away from chromatin."

We also discuss some additional possibilities, and at this stage it is difficult to know which of these are most relevant. At this stage, we also view all of these possibilities sceptically and are fully aware that it requires considerable further work to test these them. However, we felt that it was important to focus the discussion on observations made in our work. This includes the N-terminal region of INCENP which forms crystal contact with survivin and is extremely highly conserved in mammals (see sequence alignment in Figure 1A).

3. Page 12. "Importantly, T59E which mimics the mitotic phosphorylation of INCENP does not abolish DNA binding."

This statement is confusing, since the T59E mutation greatly lowers the affinity to DNA. It is not obvious why maintenance of this very weak DNA binding is "important".

We have removed this text which the referee indicates was confusing in the context of the summary paragraph on page 12.

Revised text page 12:

“Together these data support the conclusion that the interaction of the CPC with chromatin is mediated by both selective binding of histone H3 by survivin and interaction of the INCENP RRKKRR-motif with DNA. There is a slight ~1.25-fold but reproducible preference for α -satellite DNA that suggests there may be some sequence specificity to the interaction, but this requires further investigation. We then investigated the relationship between the mechanisms of chromatin binding from prophase to metaphase and CPC release from chromatin and subsequent localisation to the central spindle in anaphase.”

See also page 13:

“Importantly, INCENP^{T59E} still co-precipitated phosphorylated histone H3 in metaphase (Figure 4, T59E).”

The referee says the interaction is “very weak”, however the affinity for DNA cannot be viewed in isolation since the CPC N-terminal module makes multiple weak interactions with chromatin. This multi-valency is described in the model outlined in Figure 10. A key factor to consider is the high concentration of DNA and nucleosomes in the cell nucleus. Nucleosomes are present at approximately 80 μ M, so high affinity binding of the CPC would result in its trapping on the chromatin surface. We propose that like other chromatin bound complexes, dynamic localisation is achieved through multiple weak interactions.

4. Although I understand the possible importance of weak multivalent binding, I maintain my skepticism about the authors' claim that this weak INCENP-DNA binding is physiologically meaningful. The authors may want to discuss a possibility that other negatively charged targets on chromatin may be more relevant.

The two most abundant charged surfaces on chromatin relevant for this work are likely to be either the DNA or the surface of the nucleosomes. Nascent RNA is another possibility but since the CPC is not found at highly expressed genes that is unlikely. We have revised the text to mention the possibility that nucleosomes may be important binding sites at appropriate points:

Page 5:

“The CPC makes multiple contacts with chromatin, and MKLP2 may compete for one or more of the interactions with phosphorylated histone H3 or nucleosomes.”

Page 21:

“At present we cannot exclude the possibility that the RRKKRR motif makes additional interactions to other charged surfaces on chromatin, such as the acidic patch formed by Histone H2A/H2B on the surface of the nucleosome (Luger et al., 1997). To address all these questions will require structural and functional studies of the CPC bound to DNA-wrapped nucleosomes....”

5. Figures 6B and C. In my previous comments, I asked the authors show molecular weights of each peak determined by MALS, expecting that they had collected the data. Unfortunately, the authors did not respond to this request. Although the authors argue that tested CPI mutants do not form stable complex with MKLP2, the elution profile indicates that they indeed interact with MKLP2 by showing an apparent shift of elution peaks, and also disappearance of MKLP2 oligomers. Currently, it is not clearly stated if the middle peak represents a complex or not, but if I read the data correctly, it contains a complex of CPI mutants and MKLP2. Disclosing the MALS data will greatly help readers understand how to interpret the data. In my opinion, if proteins maintain interaction in SEC, the protein complex is not unstable. So, I feel that it is misleading to state that the CPI mutants do not form a stable complex with MKLP2. It would be better to describe that it forms a less stable complex than the wild-type does.

We have revised the figure to indicate that “no stable stoichiometric complex” is formed as the referee suggests.

[copied from 5] Currently, it is not clearly stated if the middle peak represents a complex or not, but if I read the data correctly, it contains a complex of CPI mutants and MKLP2.

We used SEC-MALS to characterise the wild type CPI80+MKLP2 complexes to show these were uniform. This data is presented in Figure 6A. SEC was then used to test if stable stoichiometric complexes could form for the mutant CPI80^{RRKKRR} and CPI58 (Figure 6B and 6C). The gels shown to the right are important for interpreting the CPI58 and CPI80^{RRKKRR} SEC traces. In these the leftward shift in the MKLP2 peak (M) relative to CPC subunits (S, B, I) can be clearly seen. The two overlapping peaks are therefore not single species, but mixtures of these subcomplexes. To make this clearer we have added marker lines to indicate the fractions enabling simpler comparison to the SDS-PAGE panels have been added. In addition, dotted line traces indicate the individual MKLP2 and CPI subcomplexes.

This is clearly described in the text on page 15:

"The minimal binding region of MKLP2⁵⁵⁷⁻⁶⁶⁸ exists in two species with masses of 25.4 and 63.7kDa (Figure 6A and S4E, yellow trace), consistent with a dimeric and a higher order oligomeric species. When mixed, CPI80 and MKLP2⁵⁵⁷⁻⁶⁶⁸ form a single species with a molecular mass of 100.3kDa (Figure 6A and S4E, blue trace), consistent with two copies of CPI80 bound to a dimer of MKLP2.

In contrast, neither the CPI80^{RRKKRR} mutant nor CPI58, which is truncated prior to the RRKKRR-motif, formed SEC stable stoichiometric complexes with MKLP2⁵⁵⁷⁻⁶⁶⁸, resulting in multiple peaks in the elution profile (Figure 6B and 6C, blue traces)."

6. Page 8, line 33. "espectively" should read "respectively".

This typo has been corrected.

Reviewer #3 (Comments to the Authors (Required)):

The revised version of the manuscript has been quite significantly improved in several aspects. The Authors have addressed the comments I have raised aside from one point.

It is quite disappointing that the Author have dismissed a comment that is quite important in this context.

"the centromere of chicken chromosomes that do not contain α -satellite and are not composed of repetitive sequences (<http://dx.doi.org/10.1016/j.devcel.2013.02.009>). If the hypothesis provided was correct, then these centromeres should accumulate less CPC or less stable one. However, the error correction mechanisms and the segregation defects are the same as for centromere with α -satellite or repetitive DNA. Therefore, the small difference observed in vitro may not be of relevance in cells.

Our work is done in human cells and is to a large extent focussed on the mechanism of CPC transport to the central spindle. The reviewer speculates about what our results might mean in chicken, but we don't feel it is appropriate for us to do the same."

The comment here is very important for the interpretation of the data. The region of INCENP involved in chromatin binding is exactly the same across species (as presented in Figure 1), the CPC dynamics and function is the same in all vertebrates and there is not a "human specific" or "chicken specific" function.

If the Authors want to make a claim, they have to take into account all the biological evidence and not only the ones that fit their hypothesis.

Since most of the chicken centromeres do not have alpha satellite, this cannot be the reader for INCENP binding to chromatin.

This was also in line with the concerns expressed by other referees.

Although the authors have down-toned the claim in the revised version and contemplate other possibilities, it should become clearer in the text.

We did not dismiss the original concerns, and carefully revised the manuscript text as the reviewer notes at the end of their own response to us – *"the authors have down-toned the claim in the revised version and contemplate other possibilities"*. However, it is important to us to address the specific point raised by the referee about interpretation of our data. Centromeric α -satellite DNA is A-T rich sequence and this feature of A-T richness is also observed for neocentromeres even if the precise DNA sequence is not conserved (reviewed in Naughton & Gilbert 2020 Exp Cell Res Vol:389). The paper from the Fukugawa and Earnshaw groups referred to by the referee is a study of neocentromere formation in chicken cells (*Chromosome engineering allows the efficient isolation of vertebrate neocentromeres*. Dev Cell (2013) 24:635-648). This work also states that both human and chicken neocentromeres form on A-T rich sequences. For ease of reference the relevant text from that work is copied below (the key portion is underlined):

"Marshall et al. (2008) previously proposed that human neocentromeres preferentially form on AT-rich sequences. We therefore determined the GC% content of the 18 neocentromeres mapped in detail. The overall GC% content of the entire chicken Z chromosome (~81 Mb) is 41%. The GC% content of 14/18 mapped neocentromeres was typically less than 41% (Figures 3A and S3). Thus, chicken neocentromeres may form preferentially on sequences with higher than average AT% content."

Thus, chicken centromeres and neocentromere, like those of human cells contain A-T rich DNA and are covered by nucleosomes. In contrast to what is implied by the referee, A-T rich sequence appears to be a general property of both centromeres and neocentromeres. If we discuss chicken centromeres and determinants for neocentromere formation in detail, then we should also discuss yeast, fly, worm and other centromeres. This requires an extensive literature review and goes beyond what is normal for the discussion section of a

manuscript. We would like to remind the reviewer and editor that our work addresses the molecular basis of the CPC transport mechanism away from chromatin in anaphase, and is not an analysis of centromere structure.

Apart from the request for this specific citation, we aren't entirely clear what the referee wants us to do. We do not say that there is an absolute requirement for specific recognition of α -satellite DNA sequences in our study. We use these A-T rich sequences for the obvious reason that they are found at human centromeres. Reviewer 3 extrapolates this to mean that the DNA binding has to be highly sequence specific, and thus our findings cannot explain CPC targeting to chicken neocentromeres. We disagree with this view for a number of reasons, as outlined above centromeres and neocentromeres tend to form at A-T rich sequence. At this stage we feel that extended discussion of this specific point would be premature, and rather focus on the key point that further studies are needed to understand binding of the CPC to DNA wrapped nucleosomes.

We have fully revised this text as follows (see also referee 1, point 4):

"There are still unanswered questions relating to this proposed mechanism. One pressing question is the selectivity of the INCENP-DNA interaction. Although both centromeres and neocentromeres are usually assembled on A-T rich DNA, there is little conservation of the underlying DNA sequence (Naughton and Gilbert, 2020). However, it has previously been reported that neocentromeres assembled away from α -satellite regions show less defined CPC targeting and altered Aurora B regulation (Bassett et al., 2010). This results in increased chromosome mis-segregation and mitotic errors (Naughton and Gilbert, 2020). Thus, while the underlying DNA sequence may not be crucial for centromere and kinetochore formation, it may facilitate recruitment of specific factors such as the CPC, which plays a crucial role in the correction of errors in chromosome alignment and segregation. Our results using A-T rich α -satellite sequences provide some evidence that INCENP RRKKRR-motif mediated DNA binding has some sequence selectivity, possibly for A-T rich DNA, which may explain these effects. However, our results do not support the view that CPC targeting is strongly sequence dependent. At present our view is that DNA binding needs to be seen in the context of the CPC-nucleosome interaction and that further studies are required to address this. In addition to the borealin-mediated interaction with nucleosomes (Abad et al., 2019), we cannot exclude the possibility that the RRKKRR motif makes additional interactions to other charged surfaces on chromatin, such as the acidic patch formed by Histone H2A/H2B on the surface of the nucleosome (Luger et al., 1997). To address all these questions will require structural and functional studies of the CPC bound to DNA-wrapped nucleosomes, and detailed comparison of CPC targeting to both centromeres and neocentromeres."

On page 11 we have added text to clarify why we selected the sequences used for analysis of DNA binding:

"We used these sequences since most centromeres and neocentromeres are characterised by the presence of AT-rich DNA, of which α -satellite is one example (Naughton and Gilbert, 2020)."

Our clearly stated hypothesis, supported by multiple different data is that CPC targeting is not due to a single feature, such as DNA sequence. We propose that both a specific nucleosome mark combined with DNA interaction is needed for chromatin and centromere targeting of the CPC. This is very different from the idea proposed by the referee that *"alpha satellite" is a "reader for INCENP binding to chromatin"*.

We would also like to note that Naughton and Gilbert write that α -satellite DNA is important for reducing errors in chromosome segregation:

"In addition, whilst neocentromeres form fully functional kinetochores and are stably propagated, they are still associated with significantly higher chromosome mis-segregation rates and mitotic errors [59,60]. Thus, the consensus α -satellite repetitive DNA may provide a safety buffer against centromeric drift [61] and reduce neocentromere instability [59,60]."

Basset and Black writing in the JCB (2010) reported that Aurora B is not correctly localised at neocentromeres (see Figure 2A and 4A in that paper). They observed a *"dilution of stable CPC (Aurora B) binding sites along pericentromeric chromatin in mitotic chromosome spread preparations"* and inappropriate silencing of Aurora B activity. The precise molecular mechanism was not determined, but may relate to our findings. However, at present this is a speculation and testing this idea would require examination of targeting of wild type and mutant CPC to neocentromeres. At present we do not have a system set up to do this.

Thus, while the underlying DNA sequence may not be crucial for centromere and kinetochore formation, it may facilitate recruitment of specific factors such as the CPC, which plays a crucial role in the correction of errors in chromosome alignment and segregation. This would be consistent with our data. However, we feel that such a discussion is too premature and we would rather focus on future work that addresses this question. We hope the editor and review respect this cautious position.